# STOCHASTIC TRAINING IS NOT NECESSARY FOR GENERALIZATION

**Jonas Geiping**
University of Siegen
jgeiping@umd.edu

**Micah Goldblum**
University of Maryland
goldblum@umd.edu

**Phillip E. Pope**
University of Maryland
pepope@cs.umd.edu

**Michael Moeller**
University of Siegen
michael.moeller@uni-siegen.de

**Tom Goldstein**
University of Maryland
tomg@umd.edu

## ABSTRACT

It is widely believed that the implicit regularization of SGD is fundamental to the impressive generalization behavior we observe in neural networks. In this work, we demonstrate that non-stochastic full-batch training can achieve comparably strong performance to SGD on CIFAR-10 using modern architectures. To this end, we show that the implicit regularization of SGD can be completely replaced with explicit regularization. Our observations indicate that the perceived difficulty of full-batch training may be the result of its optimization properties and the disproportionate time and effort spent by the ML community tuning optimizers and hyperparameters for small-batch training.

## 1 INTRODUCTION

Stochastic gradient descent (SGD) is the backbone of optimization for neural networks, going back at least as far as LeCun et al. (1998a), and SGD is the de-facto tool for optimizing the parameters of modern neural networks (Krizhevsky et al., 2012; He et al., 2015a; Brown et al., 2020). A central reason for the success of stochastic gradient descent is its efficiency in the face of large datasets – a noisy estimate of the loss function gradient is generally sufficient to improve the parameters of a neural network and can be computed much faster than a full gradient over the entire training set.

At the same time, folk wisdom dictates that small-batch SGD is not only faster but also has a unique bias towards good loss function minima that cannot be replicated with full batch gradient descent. Some even believe that stochastic sampling is the fundamental force behind the success of neural networks. These popular beliefs are linked to various properties of SGD, such as its gradient noise, fast escape from saddle points, and its uncanny ability to avoid sub-optimal local minima (Hendrik, 2017; LeCun, 2018). These properties are also attributed in varying degrees to all mini-batched first-order optimizers, such as Adam (Kingma & Ba, 2015) and others (Schmidt et al., 2020).

But why does stochastic mini-batching really aid generalization? In this work, we set out to isolate mechanisms which underlie the benefits of SGD and use these mechanisms to replicate the empirical benefits of SGD without stochasticity. In this way, we provide a counterexample to the hypothesis that stochastic mini-batching, which leads to noisy estimates of the gradient of the loss function, is fundamental for the strong generalization success of over-parameterized neural networks.

We show that a standard ResNet-18 can be trained with batch size 50K (the entire training dataset) and still achieve $95.67\%(\pm0.08)$ validation accuracy on CIFAR-10, which is comparable to the same network trained with a strong SGD baseline, provided data augmentation is used for both methods. We then extend these findings to train without (random) data augmentations, for an *entirely* non-stochastic full-batch training routine with exact computation of the full loss gradient, while still achieving over 95% accuracy. Because existing training routines are heavily optimized for small-batch SGD, the success of our experiments requires us to eschew standard training parameters in favor of more training steps, aggressive gradient clipping, and explicit regularization terms.

The existence of this example raises questions about the role of stochastic mini-batching, and by extension gradient noise, in generalization. In particular, it shows that the practical effects of such gradient noise can be captured by explicit, non-stochastic, regularization. This shows that deep learning succeeds even in the absence of mini-batched training. A number of authors have studied relatively large batch training, often finding trade-offs between batch size and model performance (Yamazaki et al., 2019; Mikami et al., 2019; You et al., 2020). However, the goal of these studies has been first and foremost to accelerate training speed (Goyal et al., 2018; Jia et al., 2018), with maintaining accuracy as a secondary goal. In this study, we seek to achieve high performance on full-batch training at all costs. Our focus is not on fast runtimes or ultra-efficient parallelism, but rather on the implications of our experiments for deep learning theory. In fact, the extremely high cost of each full-batch update makes GD far less efficient than a conventional SGD training loop.

We begin our discussion by reviewing the literature on SGD and describing various studies that have sought to explain various successes of deep learning through the lens of stochastic sampling. Then, we explain the hyper-parameters needed to achieve strong results in the full-batch setting and present benchmark results using a range of settings, both with and without data augmentation.

## 2   Perspectives on Generalization via SGD

The widespread success of SGD in practical neural network implementations has inspired theorists to investigate the gradient noise created by stochastic sampling as a potential source of observed generalization phenomena in neural networks. This section will cover some of the recent literature concerning hypothesized effects of stochastic mini-batch gradient descent (SGD). We explicitly focus on generalization effects of SGD in this work. Other possible sources of generalization for neural networks have been proposed that do not lean on stochastic sampling, for example generalization results that only require overparametrization (Neyshabur et al., 2018; Advani et al., 2020), large width (Golubeva et al., 2021), and well-behaved initialization schemes (Wu et al., 2017; Mehta et al., 2020). We will not discuss these here. Furthermore, because we wish to isolate the effect of stochastic sampling in our experiments, we fix an architecture and network hyperparameters in our studies, acknowledging that they were likely chosen because of their synergy with SGD.

**Notation.** We denote the optimization objective for training a neural network by $\mathcal{L}(x, \theta)$, where $\theta$ represents network parameters, and $x$ is a single data sample. Over a dataset $X$ of $N$ data points, $\{x_i\}_{i=1}^N$, the neural network training problem is the minimization of

$$L(\theta) = \frac{1}{N} \sum_{x \in X} \mathcal{L}(x, \theta). \tag{1}$$

This objective can be optimized via first-order optimization, of which the simplest form is descent in the direction of the negative gradient with respect to parameters $\theta$ on a batch $B$ of data points and with step size $\tau_k$:

$$\theta^{k+1} = \theta^k - \tau_k \frac{1}{|B|} \sum_{x \in B} \nabla \mathcal{L}(x, \theta^k). \tag{2}$$

Now, full-batch gradient descent corresponds to descent on the full dataset $B = X$, stochastic gradient descent corresponds to sampling a single random data point $B = \{x\} \sim X$ (with or without replacement), and mini-batch stochastic gradient descent corresponds to sampling $S$ data points $B = \{x_j\}_{j=1}^S, x_j \sim X$ at once. When sampling without replacement, the set is commonly reset after all elements are depleted.

Although stochastic gradient descent has been used intermittently in applications of pattern recognition as far back as the 90's, its advantages were debated as late as Wilson & Martinez (2003), who in support of SGD discuss its efficiency benefits (which would become much more prominent in the following years due to increasing dataset sizes), in addition to earlier ideas that stochastic training can escape from local minima, and its relationship to Brownian motion and "quasi-annealing", both of which are also discussed in practical guides such as LeCun et al. (1998b).

**SGD and critical points.** While early results from an optimization perspective were concerned with showing the effectiveness and convergence properties of SGD (Bottou, 2010), later ideas focused on the generalization of stochastic training via navigating the optimization landscape, finding global minima, and avoiding bad local minima and saddlepoints. Ge et al. (2015) show that stochastic

descent is advantageous compared to full-batch gradient descent (GD) in its ability to escape saddle points. Although the same conditions actually also allow vanilla gradient descent to avoid saddle-points (Lee et al., 2016), full-batch descent is slowed down significantly by the existence of saddle points compared to stochastically perturbed variants (Du et al., 2017). Random perturbations also appear necessary to facilitate escape from saddle points in Jin et al. (2019). It is also noted by some authors that higher-order optimization, which can alleviate these issues, does perform better in the large-batch regimes (Martens & Grosse, 2020; Yadav, 2020; Anil et al., 2021). Related works further study a *critical mini-batch size* (Ma et al., 2018; Jain et al., 2018) after which SGD behaves similarly to full-batch gradient descent (GD) and converges slowly. It is unclear though whether the analysis of sub-optimal critical points can explain the benefits of SGD, given that modern neural networks can generally be trained to reach global minima even with deterministic algorithms (for wide enough networks (Du et al., 2019)). It has been postulated that "good" minima that generalize well share geometric properties that make it likely for SGD to find them (Huang et al., 2020).

**Flatness and Noise Shapes.** One such geometric property of a global minimizer is its *flatness* (Hochreiter & Schmidhuber, 1997). Empirically, Keskar et al. (2016) discuss the advantages of small-batch stochastic gradient descent and propose that finding flat basins is a benefit of small-batch SGD: Large-batch training converges to models with both lower generalization and sharper minimizers. Although flatness is difficult to measure (Dinh et al., 2017), flatness based measures appear to be the most promising tool for predicting generalization in Jiang et al. (2019).

The analysis of such stochastic effects is often facilitated by considering the stochastic differential equation that arises for small enough step sizes $\tau$ from Eq. (2) under the assumption that the gradient noise is effectively a Gaussian random variable:

$$\mathrm{d}\theta_t = -\nabla L(\theta_t)\,\mathrm{d}t + \sqrt{\tau\Sigma_t}\,\mathrm{d}W_t, \tag{3}$$

where $\Sigma_t$ represents the covariance of gradient noise at time $t$, and $W_t$ is a Brownian motion modeling it. The magnitude of $\Sigma_t$ is inversely proportional to mini-batch size (Jastrzębski et al., 2018), and it is also connected to the flatness of minima reached by SGD in Dai & Zhu (2018) and Jastrzębski et al. (2018) if $\Sigma_t$ is isotropic. Analysis therein as well as in Le (2018) provides evidence that the step size should increase linearly with the batch size to keep the magnitude of noise fixed. However, the anisotropy of $\Sigma_t$ is strong enough to generate behavior that qualitatively differs from Brownian motion around critical points (Chaudhari & Soatto, 2018; Simsekli et al., 2019) and isotropic diffusion is insufficient to explain generalization benefits in Saxe et al. (2019).

The shape of $\Sigma_t$ is thus further discussed in Zhu et al. (2019) where anisotropic noise induced by SGD is found to be beneficial to reach flat minima in contrast to isotropic noise, Zhou et al. (2020) where it is contrasted with noise induced by Adam (Kingma & Ba, 2015), and HaoChen et al. (2020) who discuss that such parameter-dependent noise, also induced by label noise, biases SGD towards well-generalizing minima. Empirical studies in Wen et al. (2020); Wu et al. (2020) and Li et al. (2021) show that large-batch training can be improved by adding the right kind of anisotropic noise.

Notably, in all of these works, the noise introduced by SGD is in the end both *unbiased* and (mostly) Gaussian, and its disappearance in full-batch gradient descent should remove its beneficial effects. However, Eq. (3) only approximates SGD to first-order, while for non-vanishing step sizes $\tau$, Li et al. (2017) find that a second-order approximation,

$$\mathrm{d}\theta_t = -\nabla\left(L(\theta_t) + \frac{\tau}{4}||\nabla L(\theta)||^2\right)\mathrm{d}t + \sqrt{\tau\Sigma_t}\,\mathrm{d}W_t, \tag{4}$$

does include an implicit bias proportional to the step size. Later studies such as Li et al. (2020) discuss the importance of large initial learning rates, which are also not well modeled by first-order SDE analysis but have a noticeable impact on generalization.

**An explicit, non-stochastic bias?** Several of these theoretical investigations into the nature of generalization via SGD rely on earlier intuitions that this generalization effect would not be capturable by explicit regularization: Arora et al. (2019a) write that "standard regularizers may not be rich enough to fully encompass the implicit regularization brought forth by gradient-based optimization" and further rule out norm-based regularizers rigorously. Similar statements have already been shown for the generalization effects of overparametrization in Arora et al. (2018) who show that no regularizer exists that could replicate the effects of overparametrization in deep linear networks. Yet, Barrett & Dherin (2020); Smith et al. (2020b) find that the implicit regularization induced by GD and SGD

can be analyzed via backward-error analysis and a scalar regularizer can be derived. The implicit generalization of mini-batched gradient descent with batches $B \in \mathcal{B}$ can be (up to third-order terms and sampling without replacement) described explicitly by the modified loss function

$$L(\theta) + \frac{\tau}{4|\mathcal{B}|} \sum_{B \in \mathcal{B}} \left\| \frac{1}{|B|} \sum_{x \in B} \nabla \mathcal{L}(x, \theta) \right\|^2, \tag{5}$$

which simplifies for gradient descent to

$$L(\theta) + \frac{\tau}{4} \left\| \nabla L(\theta) \right\|^2, \tag{6}$$

as found in Barrett & Dherin (2020). Training with this regularizer can induce the generalization benefits of larger learning rates, even if optimized with small learning rates, and induce benefits in generalization behavior for small batch sizes when training moderately larger batch sizes. However, Smith et al. (2020b) "expect this phenomenon to break down for very large batch sizes". Related are discussions in Roberts (2018) and Poggio & Cooper (2020), who show a setting in which SGD can be shown to converge to a critical point where $\nabla \mathcal{L}(x_i, \theta) = 0$ holds separately for each data point $x$, a condition which implies that the regularizer of Eq. (5) is zero.

**Large-batch training in practice.** In response to Keskar et al. (2016), Hoffer et al. (2017) show that the adverse effects of (moderately) large batch training can be mitigated by improved hyperparameters – tuning learning rates, optimization steps, and batch normalization behavior. A resulting line of work suggests hyperparameter improvements that successively allow larger batch sizes, (You et al., 2017) with reduced trade-offs in generalization. Yet, parity in generalization between small and large batch training has proven elusive in many applications, even after extensive hyperparameter studies in De et al. (2017); Golmant et al. (2018); Masters & Luschi (2018) and Smith et al. (2020a). Golmant et al. (2018) go on to discuss that this is not only a problem of generalization in their experiments but also one of optimization during training, as they find that the number of iterations it takes to even reach low training loss increases significantly after the critical batch size is surpassed. Conversely, Shallue et al. (2019) find that training in a large-batch regime is often still possible, but this is dependent on finding an appropriate learning rate that is not predicted by simple scaling rules, and it also depends on choosing appropriate hyperparameters and momentum that may differ from their small-batch counterparts. A reduction of possible learning rates that converge reliably is also discussed in Masters & Luschi (2018), but a significant gap in generalization is observed in Smith et al. (2020a) even after grid-searching for an optimal learning rate.

Empirical studies continue to optimize hyperparameters for large-batch training with reasonable sacrifices in generalization performance, including learning rate scaling and warmup (Goyal et al., 2018; You et al., 2019a), adaptive optimizers (You et al., 2017; 2019b), omitting weight regularization on scales and biases (Jia et al., 2018), adaptive momentum (Mikami et al., 2019), second-order optimization (Osawa et al., 2019), and label smoothing Yamazaki et al. (2019). Yet, You et al. (2020) find that full-batch gradient descent cannot be tuned to reach the performance of SGD, even when optimizing for long periods, indicating a fundamental "limit of batch size". The difficulty of achieving good generalization with large batches has been linked to instability of training. As discussed in Cohen et al. (2020); Gilmer et al. (2021), training with GD progressively increases the sharpness of the objective function until training destabilizes in a sudden loss spike. Surprisingly however, the algorithm does not diverge, but quickly recovers and continues to decrease non-monotonically, while sharpness remains close to a stability threshold. This phenomenon of non-monotone, but effective training close to a stability threshold is also found in Lewkowycz et al. (2020).

## 2.1 A MORE SUBTLE HYPOTHESIS

From the above literature, we find two main advantages of SGD over GD. First, its optimization behavior appears qualitatively different, both in terms of stability and in terms of convergence speed beyond the critical batch size. Secondly, there is evidence that the implicit bias induced by large step size SGD on mini batches can be replaced with explicit regularization as derived in Eq. (4) and Eq. (5) - a bias that approximately penalizes the per-example gradient norm of every example. In light of these apparent advantages, we hypothesize that *we can modify and tune optimization hyperparameters for GD and also add an explicit regularizer in order to recover SGD's generalization performance without injecting any noise into training.* This would imply that gradient noise from

mini-batching is not necessary for generalization, but an intermediate factor; while modeling the bias of gradient noise and its optimization properties is sufficient for generalization, mini-batching by itself is not necessary and these benefits can also be procured by other means.

This hypothesis stands in contrast to possibilities that gradient noise injection is either necessary to reach state-of-the-art performance (as in Wu et al. (2020); Li et al. (2021)) or that no regularizing function exists with the property that its gradient replicates the practical effect of gradient noise (Arora et al., 2018). A "cultural" roadblock in this endeavor is further that existing models and hyperparameter strategies have been extensively optimized for SGD, with a significant number of hours spent improving performance on CIFAR-10 for models trained with small batch SGD, which begets the question whether these mechanisms are by now self-reinforcing?

## 3 FULL-BATCH GD WITH RANDOMIZED DATA AUGMENTATION

We now investigate our hypothesis empirically, attempting to set up training so that strong generalization occurs even without gradient noise from mini-batching. We will thus compare *full-batch* settings in which the gradient of the full loss is computed every iteration and *mini-batch* settings in which a noisy estimate of the loss is computed. Our central goal is to *reach good full-batch performance without resorting to gradient noise*, via mini-batching or explicit injection. Yet, we will occasionally make remarks regarding *full-batch* in practical scenarios outside these limitations.

For this, we focus on a well-understood case in the literature and train a ResNet model on CIFAR-10 for image classification. We consider a standard ResNet-18 (He et al., 2015a; 2019) with randomly initialized linear layer parameters (He et al., 2015b) and batch normalization parameters initialized with mean zero and unit variance, except for the last in each residual branch which is initialized to zero (Goyal et al., 2018). This model and its initialization were tuned to reach optimal performance when trained with SGD. The default random CIFAR-10 data ordering is kept as is.

We proceed in several stages from baseline experiments using standard settings to specialized schemes for full-batch training, comparing stochastic gradient descent performance with full-batch gradient descent. Over the course of this and the next section we first examine full-batch training with standard data augmentations, and later remove randomized data augmentations from training as well to evaluate a completely noise-less pipeline.

### 3.1 BASELINE SGD

We start by describing our baseline setup, which is well-tuned for SGD. For the entire Section 3, every image is randomly augmented by horizontal flips and random crops after padding by 4 pixels.

**Baseline SGD:** For the SGD baseline, we train with SGD and a batch size of 128, Nesterov momentum of 0.9 and weight decay of 0.0005. Mini-batches are drawn randomly without replacement in every epoch. The learning rate is warmed up from 0.0 to 0.1 over the first 5 epochs and then reduced via cosine annealing to 0 over the course of training (Loshchilov & Hutter, 2017). The model is trained for 300 epochs. In total, $390 \times 300 = 117,000$ update steps occur in this setting.

With these hyperparameters, mini-batch SGD (sampling without replacement) reaches a validation accuracy of $95.70\%(\pm 0.05)$, which we consider a very competitive modern baseline for this architecture. Mini-batch SGD provides this strong baseline largely independent from the exact flavor of mini-batching, more details can be found in the appendix.

With the same settings, we now switch to full-batch gradient descent. We replace the mini-batch updates by full batches and accumulate the gradients over all mini-batches. To rule out confounding effects of batch normalization, batch normalization is still computed over blocks of size 128 (Hoffer et al., 2017), although the assignment of data points to these blocks is kept fixed throughout training so that no stochasticity is introduced by batch normalization. In line with literature on large-batch training, applying full-batch gradient descent with these settings reaches a validation accuracy of only $75.42\%(\pm 00.13)$, yielding a $\sim 20\%$ gap in accuracy between SGD and GD. In the following experiments, we will close the gap between full-batch and mini-batch training. We do this by eschewing common training hyper-parameters used for small batches, and re-designing the training pipeline to maintain stability without mini-batching.

| Experiment | Mini-batching | Epochs | Steps | Modifications | Val. Acc.% |
|---|---|---|---|---|---|
| Baseline SGD | ✓ | 300 | 117,000 | - | 95.70($\pm$0.11) |
| SGD regularized | ✓ | 300 | 117'000 | reg | 95.81($\pm$0.18) |
| Baseline FB | ✗ | 300 | 300 | - | 75.42($\pm$0.13) |
| FB train longer | ✗ | 3000 | 3000 | - | 87.36($\pm$1.23) |
| FB clipped | ✗ | 3000 | 3000 | clip | 93.85($\pm$0.10) |
| FB regularized | ✗ | 3000 | 3000 | clip+reg | 95.54($\pm$0.09) |
| FB strong reg. | ✗ | 3000 | 3000 | clip+reg+bs32 | 95.68($\pm$0.09) |
| FB in practice | ✗ | 3000 | 3000 | clip+reg+bs32+shuffle | 95.91($\pm$0.14) |

Table 1: Validation accuracies on the CIFAR-10 validation set for each experiment with data augmentations considered in Section 3. All validation accuracies are averaged over 5 runs.

## 3.2 STABILIZING TRAINING

Training with huge batches leads to unstable behavior. As the model is trained close to its *edge of stability* (Cohen et al., 2020), we soon encounter spike instabilities, where the cross entropy objective $\mathcal{L}(\theta)$ suddenly increases in value, before quickly returning to its previous value and improving further. While this behavior can be mitigated with small-enough learning rates and aggressive learning rate decay (see supp. material), small learning rates also mean that the training will firstly make less progress, but secondly also induce a smaller implicit gradient regularization, i.e. Eq. (6). Accordingly, we seek to reduce the negative effects of instability while keeping learning rates from vanishing. In our experiments, we found that very gentle warmup learning rate schedules combined with aggressive *gradient clipping* enables us to maintain stability with a manageable learning rate.

**Gentle learning rate schedules.** Because full-batch training is notoriously unstable, the learning rate is now warmed up from 0.0 to 0.4 over 400 steps (each step is now an epochs) to maintain stability, and then decayed by cosine annealing (with a single decay without restarts) to 0.1 over the course of 3000 steps/epochs.

The initial learning rate of $0.4$ is not particularly larger than in the small-batch regime, and it is extremely small by the standards of a linear scaling rule (Goyal et al., 2018), which would suggest a learning rate of 39, or even a square-scaling rule (Hoffer et al., 2017), which would predict a learning rate of $1.975$ when training longer. As the size of the full dataset is certainly larger than any critical batch size, we would not expect to succeed in fewer steps than SGD. Yet, the number of steps, 3000, is simultaneously huge, when measuring efficiency in passes through the dataset, and tiny, when measuring parameter update steps. Compared to the baseline of SGD, this approach requires a ten-fold increase in dataset passes, but it provides a 39-fold decrease in parameter update steps. Another point of consideration is the effective learning rate of Li & Arora (2019). Due to the effects of weight decay over 3000 steps and limited annealing, the effective learning rate is not actually decreasing during training.

Training with these changes leads to full-batch gradient descent performance of $87.36\%(\pm1.23)$, which is a $12\%$ increase over the baseline, but still ways off from the performance of SGD. We summarize validation scores in Table 1 as we move across experiments.

**Gradient Clipping.** We clip the gradient over the entire dataset to have an $\ell^2$ norm of at most $0.25$ before updating parameters.

Training with all the previous hyperparameters and additional clipping obtains a validation accuracy of $93.85(\pm0.10)$. This is a significant increase of $6\%$ over the previous result due to a surprisingly simple modification, even as other improvements suggested in the literature (label smoothing (Yamazaki et al., 2019), partial weight decay (Jia et al., 2018), adaptive optimization (You et al., 2017), sharpness-aware minimization (Foret et al., 2021) fail to produce significant gains, see appendix).

Gradient clipping is used in some applications to stabilize training (Pascanu et al., 2013). However in contrast to its usual application in mini-batch SGD, where a few batches with high gradient contributions might be clipped in every epoch, here the entire dataset gradient is clipped. As such, the method is not a tool against heavy-tailed noise (Gorbunov et al., 2020), but it is effectively a limit on the maximum distance moved in parameter space during a single update. Because clipping simply changes the size of the gradient update but not its direction, clipping is equivalent to choosing

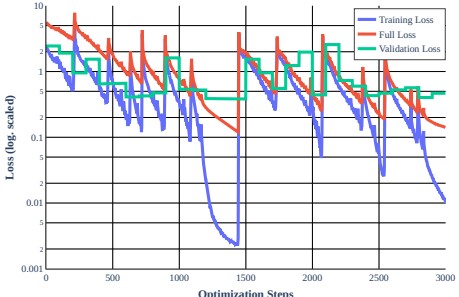 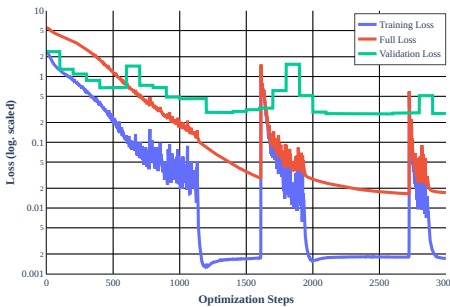

Figure 1: Cross-Entropy Loss on the training and validation set and full loss (including weight decay) during training for full-batch gradient descent. Left: training as described in Section 3.2 without clipping, right: with gradient clipping. Validation computed every 100 steps.

a small learning rate when the gradient is large. Theoretical analysis of gradient clipping for GD in Zhang et al. (2019b) and Zhang et al. (2020) supports these findings, where it is shown that clipped descent algorithms can converge faster than unclipped algorithms for a class of functions with a relaxed smoothness condition. Clipping also does not actually repress the spike behavior entirely. To do so would require a combination of even stronger clipping and reduced step sizes, but the latter would reduce both training progress and regularization via Eq. (6).

### 3.3    Bridging the gap with Explicit Regularization

Finally, there is still the bias of mini-batch gradient descent towards solutions with low gradient norm per batch described in Eq. (4) and Eq. (5). This bias, although a 2nd-order effect, is noticeable in our experiments. We can replicate this bias as an explicit regularizer via Eq. (5). However, computing exact gradients of this regularizer directly is computationally expensive due to the computation of repeated Hessian-vector products in each accumulated batch, especially within frameworks without forward automatic differentiation which would allow for the method of Pearlmutter (1994) for fast approximation of the Hessian. As such, we approximate the gradient of the regularizer through a finite-differences approximation and compute

$$\nabla \frac{1}{2} \left\| \nabla \mathcal{L}(x, \theta) \right\|^2 \approx \frac{\nabla \mathcal{L}\left(x, \theta + \varepsilon \nabla \mathcal{L}(x, \theta)\right) - \nabla \mathcal{L}(x, \theta)}{\varepsilon}. \tag{7}$$

This approximation only requires one additional forward-backward pass, given that $\nabla \mathcal{L}(x, \theta)$ is already required for the main loss function. Its accuracy is similar to a full computation of the Hessian-vector products (see supplementary material). In all experiments, we set $\varepsilon = \frac{0.01}{||\nabla \mathcal{L}(x, \theta)||}$, similar to (Liu et al., 2018a). To compute Eq. (5), the same derivation is applied for averaged gradients $\frac{1}{|B|} \sum_{x \in B} \nabla \mathcal{L}(x, \theta)$.

**Gradient Penalty.** We regularize the loss via the gradient penalty in Eq. (5) with coefficient $\frac{\alpha \tau_k}{4}$. We set $\alpha = 1.0$ for these experiments.

We use this regularizer entirely without sampling, computing it over the fixed mini-batch blocks $B \in \mathcal{B}$, already computed for batch normalization, which are never shuffled. We control the strength of the regularization via a parameter $\alpha$. Note that this regularizer can be computed in parallel across all batches in the dataset. Theoretical results from Smith et al. (2020b) do not guarantee that the regularizer can work in this setting, especially given the relatively large step sizes we employ. However, the regularizer leads to the direct effect that not only $||\nabla L(\theta)||$ is small after optimization, but also $|| \sum_{x \in B} \nabla \mathcal{L}(x, \theta)||$, i.e. the loss on each mini-batch. Intuitively, the model is optimized so that it is still optimal when evaluated only on subsets of the training set (such as these mini-batches).

Applying this regularizer on top of clipping and longer training leads to a validation accuracy of $94.70\%(\pm 0.17)$ for a regularizer accumulated over the default batch size of $|B| = 128$. This can be further increased to $94.97(\pm 0.05)$ if the batch size is reduced to 32 (reducing the SGD batch size in the same way does not lead to additional improvement, see supp. material). Reducing $|B|$ is a beneficial effect as the regularizer Eq. (5) is moved closer to a direct penalty on the per-example gradient norms of Poggio & Cooper (2020), yet computational effort increases proportionally.

| Experiment | ResNet-18 | ResNet-50 | Resnet-152 | DenseNet-121 | VGG-16 |
|---|---|---|---|---|---|
| Baseline SGD | 95.70($\pm$0.11) | 95.65($\pm$0.18) | 95.80($\pm$0.15) | 95.80($\pm$0.25) | 94.42($\pm$0.89) |
| Baseline FB | 75.42($\pm$0.13) | 55.03($\pm$3.89) | 77.26($\pm$0.28) | 77.26($\pm$0.28) | 46.95($\pm$19.51) |
| FB train longer | 87.36($\pm$1.23) | 85.12($\pm$0.00) | 91.67($\pm$0.35) | 89.56($\pm$0.00) | 89.36($\pm$1.19) |
| FB clipped | 93.85($\pm$0.10) | 92.55($\pm$0.36) | 92.48($\pm$0.46) | 92.59($\pm$0.16) | 92.55($\pm$0.21) |
| FB regularized | 95.54($\pm$0.09) | 95.84($\pm$0.09) | 95.98($\pm$0.12) | 95.92($\pm$0.09) | 93.86($\pm$0.18) |
| FB strong reg. | 95.68($\pm$0.09) | 96.06($\pm$0.04) | 96.21($\pm$0.12) | 96.08($\pm$0.13) | 93.91($\pm$0.17) |
| FB in practice | 95.91($\pm$0.14) | 96.50($\pm$0.13) | 96.37($\pm$0.56) | 96.43($\pm$0.10) | 94.44($\pm$0.07) |

Table 2: Validation accuracies on the CIFAR-10 validation set for each of the experiments with data augmentations considered in Section 3 for multiple modern CNNs.

**Double the learning rate.** We again increase the initial learning rate, now to 0.8 at iteration 400, which then decays to 0.2 over the course of 3000 steps/epochs.

This second modification of the learning rate is interestingly only an advantage after the regularizer is included. Training with this learning rate and clipping, but without the regularizer (i.e. as in Section 3.2), reduces that accuracy slightly to 93.75%($\pm$0.13). However, the larger learning rate does improve the performance when the regularizer is included, reaching 95.54($\pm$0.09) if $|B| = 128$ and 95.68($\pm$0.09) if $|B| = 32$, which is finally fully on par with SGD.

Overall, we find that after all modifications, both full-batch (with random data augmentations) and SGD behave similarly, achieving significantly more than 95% validation accuracy. Figure 4 visualizes the loss landscape around the found solution throughout these changes. Noticeably both clipping and gradient regularization correlate with a flatter landscape.

**Remark** (The Practical View). *Throughout these experiments with full-batch GD, we have decided not to shuffle the data in every epoch to rule out confounding effects of batch normalization. If we turn on shuffle again, we reach 95.91%($\pm$0.14) validation accuracy (with separate runs ranging between 96.11% and 95.71%), which even slightly exceeds SGD. This is the practical view, given that shuffling is nearly for free in terms of performance, but of course potentially introduces a meaningful source of gradient noise - which is why it is not our main focus.*

Furthermore, to verify that this behavior is not specific to the ResNet-18 model considered so far, we also evaluate related vision models with exactly the same hyperparameters. Results are found in Table 2, where we find that our methods generalize to the ResNet-50, ResNet-152 and a DenseNet-121 without any hyperparameter modification. For VGG-16, we do make a minimal adjustment and increase the clipping to 1.0, as the gradients there are scaled differently, to reach parity with SGD.

## 4 FULL-BATCH GD IN THE TOTALLY NON-STOCHASTIC SETTING

A final question remains – if the full-batch experiments shown so far work to capture the effect of mini-batch SGD, what about the stochastic effect of random data augmentations on gradient noise? It is conceivable that the generalization effect is impacted by the noise variance of data augmentations. As such, we repeat the experiments of the last section in several variations.

**No Data Augmentation.** If we do not apply any data augmentations and repeat previous experiments, then GD with clipping and regularization at 89.17%, substantially beats SGD with default hyperparameters at 84.32%($\pm$1.12) and nearly matches SGD with newly tuned hyperparameters at 90.07($\pm$0.48), see Table 3. Interestingly, not only does the modified GD match SGD, the modified GD is even more stable, as it works well with the same hyperparameters as described in the previous section, and we must tune SGD even though it benefits from the same regularization implicitly.

**Enlarged CIFAR-10** To analyze both GD and SGD in a setting were they enjoy the benefits of augmentation, but without stochasticity, we replace the random data augmentations with a fixed increased CIFAR-10 dataset. This dataset is generated by sampling $N$ random data augmentations for each data point before training. These samples are kept fixed during training and never resampled, resulting in an $N$-times larger CIFAR-10 dataset. This dataset contains the same kind of variations that would appear through data augmentation, but is entirely devoid of stochastic effects on training.

If we consider this experiment for a $10\times$ enlarged CIFAR-10 dataset, then we do recover a value of 95.11%. Note that we present this experiments only because of its implications for deep learning

| Experiment | Fixed Dataset | Mini-batching | Steps | Modifications | Val. Acc. |
|---|---|---|---|---|---|
| Baseline SGD | CIFAR-10 | ✓ | $117,000$ | - | $84.32(\pm1.12)$ |
| Baseline SGD* | CIFAR-10 | ✓ | $117,000$ | - | $90.07(\pm0.48)$ |
| FB strong reg. | CIFAR-10 | ✗ | $3000$ | clip+reg+bs32 | $89.17(\pm0.24)$ |
| Baseline SGD | $10\times$ CIFAR-10 | ✓ | $117,000$ | - | $95.20(\pm0.09)$ |
| FB train longer | $10\times$ CIFAR-10 | ✗ | $3000$ | - | $88.44(-)$ |
| FB strong reg. | $10\times$ CIFAR-10 | ✗ | $3000$ | clip+reg+bs32 | $95.11(-)$ |

Table 3: Validation accuracies on the CIFAR-10 validation set for *fixed* versions of the dataset with no random data augmentations in Section 4. Hyperparameters fixed from the previous section except for SGD marked*, where the learning rate is doubled to 0.2 for a stronger baseline.

theory; computing the gradient over the enlarged CIFAR-10 is $N$-times as expensive, and there are additional training expenses incurred through increased step numbers and regularization. For this reason we do not endorse training this way as a practical mechanism. Note that SGD still have an advantage over $10\times$ CIFAR – SGD sees 300 augmented CIFAR-10 datasets, once each, over its 300 epochs of training. If we take the same enlarged CIFAR-10 dataset and train SGD by selecting one of the 10 augmented versions in each epoch, then SGD reaches $95.20\%(\pm0.09)$.

Overall, we find that we can reach more than $95\%$ validation accuracy entirely without stochasticity, after disabling gradient noise induced via mini-batching, shuffling as well as via data augmentations. The gains of $\sim 6\%$ compared to the setting without data augmentations are realized only through the increased dataset size. This shows that noise introduced through data augmentations does not appear to influence generalization in our setting and is by itself also not necessary for generalization.

## 5 DISCUSSION & CONCLUSIONS

SGD, which was originally introduced to speed up computation, has become a mainstay of neural network training. The hacks and tricks at our disposal for improving generalization in neural models are the result of millions of hours of experimentation in the small batch regime. For this reason, it should come as no surprise that conventional training routines work best with small batches. The heavy reliance of practitioners on small batch training has made stochastic noise a prominent target for theorists, and SGD is and continues to be the practical algorithm of choice, but the assumption that stochastic mini-batching by itself is the unique key to reaching the impressive generalization performance of popular models may not be well founded.

In this paper, we show that full-batch training matches the performance of stochastic small-batch training for a popular image classification benchmark. We observe that (i) with randomized augmentations, full-batch training can match the performance of even a highly optimized SGD baseline, reaching $95.67\%$ for a ResNet-18 on CIFAR-10, (ii) without any form of data augmentation, fully non-stochastic training beats SGD with standard hyper-parameters, matching it when optimizing SGD hyperparameters, and (iii) after a $10\times$ fixed dataset expansion, full-batch training with no stochasticity exceeds $95\%$, matching SGD on the same dataset. Nonetheless, our training routine is highly inefficient compared to SGD (taking far longer run time), and stochastic optimization remains a great practical choice for practitioners in most settings.

The results in this paper focus on commonly used vision models. While the scope may seem narrow, the existence of these counter-examples is enough to show that stochastic mini-batching, and by extension gradient noise, is not required for generalization. It also strongly suggests that any theory that relies exclusively on stochastic properties to explain generalization is unlikely to capture the true phenomena responsible for the success of deep learning. Stochastic sampling has become a focus of the theory community in efforts to explain generalization. However, experimental evidence in this paper and others suggests that strong generalization is achievable with large or even full batches in several practical scenarios. If stochastic regularization does indeed have benefits in these settings that cannot be captured through non-stochastic, regularized training, then those benefits are just the cherry on top of a large and complex cake.

## ETHICS STATEMENT

We foresee no direct social impact of this work at the moment.

## REPRODUCIBILITY STATEMENT

We detail all hyperparameters in the main body and provide all additional details in Appendix A. Our open source implementation can be found at `https://github.com/JonasGeiping/fullbatchtraining` and contains the exact implementation with which these results were computed and we further include all necessary scaffolding we used to run distributed experiments on arbitrarily many GPU nodes, as well as model checkpointing to run experiments on only a single machine with optional GPU. Overall, we thus believe that our experimental evaluation is accessible and reproducible for a wide range of interested parties.

## FUNDING STATEMENT

This research was made possible by the OMNI cluster of the University of Siegen which contributed a notable part of its GPU resources to the project and we thank the Zentrum für Informations- und Medientechnik of the University of Siegen for their support. We further thank the University of Maryland Institute for Advanced Computer Studies for additional resources and support through the Center for Machine Learning cluster. This research was overall supported by the universities of Siegen and Maryland and by the ONR MURI program, AFOSR MURI Program, and the National Science Foundation Division of Mathematical Sciences.

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

# A EXPERIMENTAL SETUP

## A.1 EXPERIMENTAL DETAILS

As mentioned in the main body, all experiments are evaluated on the CIFAR-10 dataset. The data is normalized per color channel. When augmented, the augmentations are random horizontal flips and random crops of size $32 \times 32$ after zero-padding by 4 pixels in both spatial dimensions. For the experiments with a fixed $N \times$ CIFAR-10, the data is fully written to a database (LMDB) in $N$ rounds, to guarantee that the dataset is fixed. The same fixed dataset is used for all experiments using that dataset. The ResNet-18 model used for most of the experiments is the default model as described in

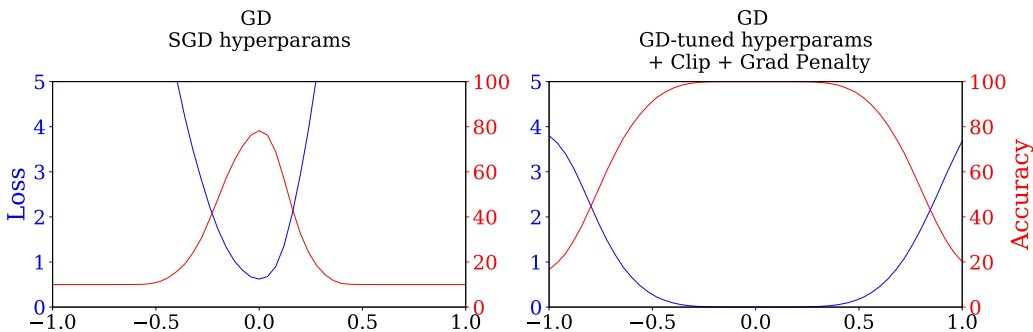

Figure 2: One-dimensional loss landscapes (random direction) of models trained with gradient descent. Default full-batch gradient descent (left) produces sharp models that do not train and generalize well, yet it can be modified to converge to flatter minima with longer training, gradient clipping and appropriate regularization (right).

He et al. (2015a; 2019) with the usual CIFAR-10 adaption of replacing the ImageNet stem ($7 \times 7$ convolution and max-pooling) with a $3 \times 3$ convolution. All experiments run in `float32` precision.

As mentioned further, the batch size during gradient accumulation is 128 if not otherwise mentioned. Gradients are averaged over all machines and batches using a running mean. The optimizer is always gradient descent with Nesterov momentum ($m = 0.9$) with learning rates as specified in the main body. Weight decay is applied to all layers. The learning rate in the basic full-batch setting with 3000 steps decays from 0.4 to 0.0 (after the 400 step linear warmup from 0.0 to 0.4) via cosine annealing (with a single cycle as in (He et al., 2019)) over 4000 ticks, so that 0.1093 is reached at the final iterate 3000 (The algorithm could be run for the additional 1000 steps to anneal to 0, but we did not find that this hurt or hindered generalization performance and as such iterate only up to 3000 steps for efficiency reasons). For the initial learning rate of 0.8 used later, this corresponds to the same schedule with warmup, but starting from 0.8 at iteration 400, which decays to 0.2187 at iteration 3000. The gradient clipping is computed based on the $\ell^2$ norm of the fully accumulated gradient vector (after addition of regularizer gradients if applicable) and the gradient vector is divided by this norm value with a fudge factor of 1e-6, if the target norm value is exceeded. This is also the PyTorch (Paszke et al., 2017) gradient clipping fudge factor. Batch normalization statistics are accumulated sequentially. If multiple GPUs are used then these accumulated statistics are averaged over all machines before each validation.

Gradient regularization as described in the main body via forward differences approximation is implemented by in-place addition of the already computed batch gradient to the model parameters with differential length $\varepsilon = 0.01/||\frac{1}{|B|}\sum_{i \in B}\mathcal{L}(x_i, \theta)||$ as suggested in (Liu et al., 2018a). The gradient at this offset location is computed through automatic differentiation as usual and the finite difference of both gradients is added to the loss gradient with the factor $\frac{\alpha \tau_k}{4}$. Afterwards the original values of the model parameters in-place are restored exactly from a copy.

All reported statistics are based on averaged results over 4-5 trials in all cases where standard deviation is reported. Numbers without standard deviation correspond to single runs. In a few edge cases where spike behavior such as in Figure 1 is seen at the final iteration 3000, and training and validation loss are accordingly large, we report maximal validation accuracy, although the value of validation accuracy at minimal full training loss or at the last iterate with gradient norm less than the clip value would also be a sensible fallbacks.

The loss landscape visualizations in Figure 2 and Figure 4 are computed by sampling the loss landscape of a fixed model (with batch normalization in evaluation mode) in a fixed random direction, which is drawn by filter normalization as described in Li et al. (2018) in a single dimension.

We provide the code to repeat these experiments with our PyTorch implementation at `github. com/JonasGeiping/fullbatchtraining`.

## A.2 COMPUTATIONAL SETUP

This experimental setup is implemented in PyTorch (Paszke et al., 2017), version 1.9. All experiments are run on an internal SLURM cluster of $4 \times 4 + 8$ NVIDIA Tesla V100-PCIE-16GB GPUs. Jobs in the data-augmented setting were mainly run on the single GPUs one at a time, whereas the $N \times$ CIFAR-10 jobs and in the fixed dataset section and jobs with gradient regularization were run distributed over $2 \times 4$ GPUs. In both settings, this amounts to 16-32 GPU hours (depending on hyperparameters, especially gradient regularization) of computation time for each experiment, times $N$ for the repeated $N \times$ CIFAR-10 variants.

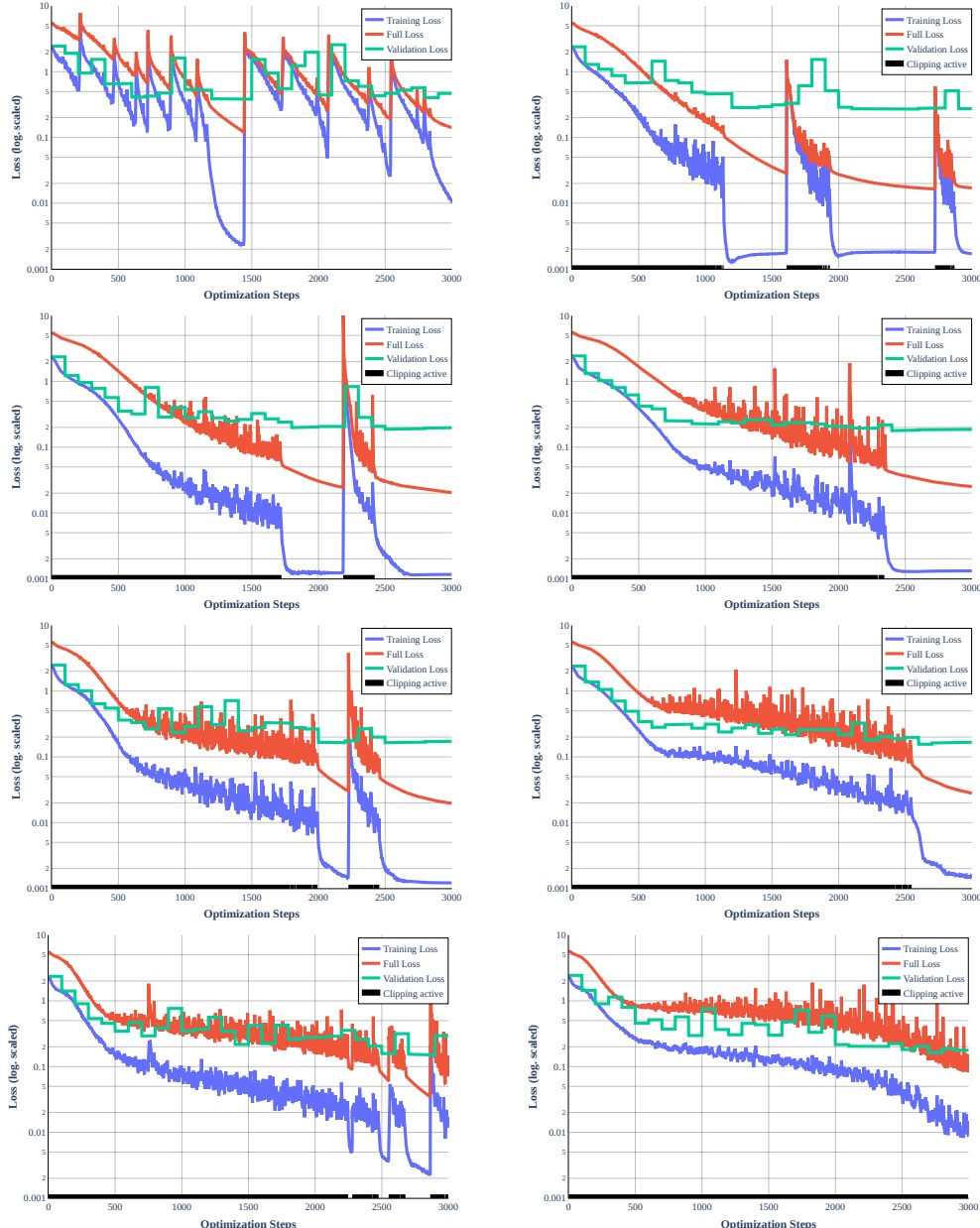

Figure 3: Cross-Entropy Loss on the training and validation set and full loss (including weight decay) during training for full-batch gradient descent. Clipped steps are marked in black. Validation computed every 100 steps. From top to bottom: Top: training as described in Section 3.2 with and without clipping. All other rows: training with gradient regularization: *FB regularized*) on the left and *FB strong reg.* on the right. Second from the top: Training with lr=0.4. Third from the top: Training with lr=0.8 (*this is the main setting investigated in this work*). Bottom: Training with lr=1.6.

## B    FURTHER RELATED WORK

The phenomenon of effective non-existence of sub-optimal local minima is itself puzzling as sub-optimal local minima do exist and can be found by specialized optimization techniques (Yun et al., 2018; Goldblum et al., 2020), but they are not found by first-order descent methods with standard initialization.

The update equation Eq. (2) is often analyzed as an update of the full-batch gradient that is contaminated by gradient noise arising from the stochastic mini-batch sampling, in this setting gradient noise is defined via:

$$\theta^{k+1} = \theta^k - \tau_k \underbrace{\nabla L(\theta^k)}_{\text{full loss gradient}} + \tau_k \left( \underbrace{\frac{1}{|X|} \sum_{x \in X} \nabla \mathcal{L}(x, \theta^k) - \frac{1}{|B|} \sum_{x \in B} \nabla \mathcal{L}(x, \theta^k)}_{\text{gradient noise } g_k} \right). \tag{8}$$

Analysis of flatness through other means, such as dynamical system theory (Wu et al., 2018; Hu et al., 2018), also derives stability conditions for SGD and GD, where among all possible global minima, SGD both converges to flatter minima than GD and also can escape from sharp minima. Xing et al. (2018) analyze SGD and GD empirically in response to the aforementioned theoretical findings about noise shape, finding that both algorithms (without momentum) significantly differ in their exploration of the loss landscape and that the structure of the noise induced by SGD is closely related to this behavior. Yin et al. (2018) introduce gradient diversity as a measure of the effectiveness of SGD:

$$\Delta_D(\theta) = \frac{\sum_{x \in X} ||\nabla \mathcal{L}(x_i, \theta)||^2}{N^2 ||\nabla L(\theta)||^2}, \tag{9}$$

which works well up to a critical batch size proportional to $\Delta_D(\theta)$. Crucially gradient diversity is a ratio of per-example gradient norms to the full gradient norm. This relationship is also investigated as gradient coherence in Chatterjee (2020) as it depends on the amount of alignment of these gradient vectors. Additional analysis of SGD as a diffusion process is facilitated in Xie et al. (2020).

Another angle to a theoretical understandings of the effects of SGD is the theoretical analysis based on convex optimization theory proposed in a series of works (Dauber et al., 2020; Bassily et al., 2020; Amir et al., 2021a;b). This line of work shows that convex loss functions can be constructed on which SGD converges to optimal generalization error orders of magnitude fast than full-batch GD.

As mentioned, *critical mini-batch size* are studied in (Ma et al., 2018; Jain et al., 2018) after which SGD behaves similarly to full-batch gradient descent (GD) and converges slowly. This idea of a critical batch size is echoed for noisy quadratic models in Zhang et al. (2019a), and an empirical measure of critical batch size is proposed in McCandlish et al. (2018). There are also hypotheses (HaoChen et al., 2020) that GD necessarily overfits at sub-optimal minima as it trains in the linearized neural tangent kernel regime of Jacot et al. (2018); Arora et al. (2019b). Additional regularization, for example via clever weight decay scheduling as in Xie et al. (2021) improves training in large(r)-batch settings, possibly by alleviating such overfitting effects.

## C    ABLATION STUDIES

### C.1    ADDITIONAL VARIATIONS

We add additional information for Table 1 in Table 4, listing training loss as well as full loss (which includes regularizations). Table 5 contains several ablation studies centered around number of steps and learning rate. We visualize the experiments from Table 1 as well as variations in learning rate in Fig. 3.

Overall, we find that the hyperparameters of the final variant (gradient descent with both clipping and regularization) are surprisingly stable. For example, we find that any learning rate within $[0.4, 1.6]$ ends up at a generalization performance around $95\%$ $[94.70, 95.69]$ which is a large stable interval,

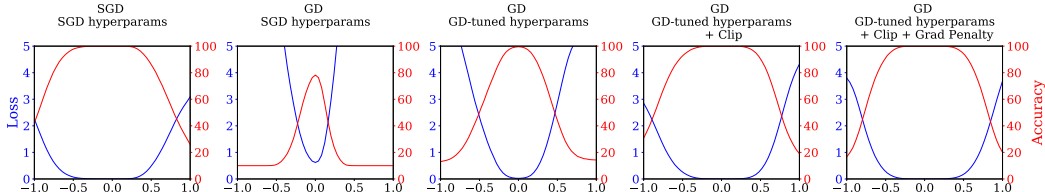

Figure 4: One-dimensional loss landscapes visualizations (random direction) of models trained with gradient descent, going from SGD (left) to GD with successive modifications (right).

compared even to baseline stochastic gradient descent. Relatedly, although we do decay the learning rate slightly (i.e. from $0.8$ to $0.2$), this leads to only a minor benefit and models generalizing almost as well can also be found with a fixed learning rate. For clipping, we mostly stuck to the gradient norm value of $0.25$ suggested by related literature for ResNet architectures (although any clip value in $[0.25, 1]$ appeared decent) and did not spend significant efforts tuning it further (aside from experiments with VGG, where we find that $0.25$ was indeed too small to allow the model to make sufficient training progress and where we upped the clipping value to $1.0$ as described in the main body). A coefficient $\alpha$ for the strength of the regularizer was introduced in the main body, yet from a brief exploratory analysis, we found that the value of $\alpha = 1$ suggested by theory was indeed optimal (Note that this is implemented as $0.5$ in the code to account for the factor $\frac{1}{2}$ in Eq. (7)). The finite differences approximation via $0.01/||\nabla L(\theta^k)||$ from Liu et al. (2018a) worked well out of the box, although we found similar results for $\varepsilon \in [5e-2, 1e-3]$. We also experimented with a higher-precision finite difference (see below). Several other parameters, such as batch norm momentum, Nesterov momentum, weight decay and model initialization were adopted directly from the SGD ResNet18 baseline model and not investigated.

| Experiment | Steps | Modifications | Val. Acc.% | Train Loss | Full Loss |
|---|---|---|---|---|---|
| Baseline SGD | 117'000 | - | 95.70($\pm$0.11) | 0.0017($\pm$4.43e$-$5) | 0.0976($\pm$5.02e$-$4) |
| Baseline FB | 300 | - | 75.42($\pm$0.13) | 0.521($\pm$3.71e$-$2) | - |
| FB train longer | 3000 | - | 87.36($\pm$1.23) | 0.0353($\pm$2.17e$-$2) | 0.0415($\pm$2.22e$-$2) |
| FB clipped | 3000 | clip | 93.85($\pm$0.10) | 0.00173($\pm$3.26e$-$5) | 0.0170($\pm$1.10e$-$4) |
| FB regularized | 3000 | clip+reg | 95.54($\pm$0.09) | 0.00161($\pm$6.91e$-$4) | 0.0259($\pm$4.01e$-$3) |
| FB strong reg. | 3000 | clip+reg+bs32 | 95.68($\pm$0.09) | 0.00158($\pm$1.25e$-$4) | 0.0381($\pm$1.02e$-$2) |
| FB in practice | 3000 | clip+reg+bs32+shuffle | 95.91($\pm$0.14) | 0.00236($\pm$2.01e$-$4) | 0.1090($\pm$7.60e$-$2) |

Table 4: Experiments considered in Table 1 with additional information in training loss (cross entropy loss on the CIFAR-10 training set) and full loss, including regularizations. Validation accuracies on the CIFAR-10 validation set for each experiment with data augmentations considered in Section 3. Best viewed on screen.

## C.2 VARIOUS TYPES OF MINIBATCHING

As described in the main body, mini-batch SGD (sampling without replacement) reaches a validation accuracy of $95.70\%(\pm0.05)$, which we consider a very competitive modern baseline for this architecture. Mini-batch SGD provides this strong baseline largely independent from the exact flavor of mini-batching as can be seen in Table 6, reaching the same accuracy when sampling with replacement. In both cases the gradient noise induced by random mini-batching leads to strong generalization. If batches are sampled without replacement and in the same order every epoch, i.e. without shuffling in every epoch, then mini-batching still provides its generalization benefit. The apparent discrepancy between both versions of shuffling is not actually a SGD effect, but shuffling benefits the batch normalization layers also present in the ResNet-18. This can be seen by replacing batch norm with group normalization (Wu & He, 2018), which has no dependence on batching. Without shuffling we find $94.44\%(\pm0.21)$ for SGD and with shuffling $94.55\%(\pm0.16)$ for group normalized ResNets; a difference of less than $1\sigma$. Overall all of these variations of mini-batched stochastic gradient descent lead to strong generalization after training. As a validation of previous work, we also note that the gap between SGD and GD is not easily closed by injecting simple forms of gradient noise, such as additive or multiplicative noise, as can also be seen in Table 6.

| Experiment | Steps | Modifications | Val. Acc.% |
|---|---|---|---|
| SGD regularized | 117'000 | reg | 95.81%($\pm$0.18)(5) |
| SGD all changes | 117'000 | clip+reg | 93.23%($-$) |
| SGD all changes strong reg | 117'000 | clip+reg+bs32 | 84.35%($-$) |
| FB clipped | 300 | clip | 84.40%($-$) |
| FB strong reg. | 300 | clip+reg+bs32 | 86.28%($\pm$0.01)(2) |
| FB strong reg. long | 40'000 | reg+bs32+lr=0.08 | 92.19%($-$) |
| FB strong reg | 6000 | clip+reg+bs32 | 95.68%($\pm$0.10)(2) |
| FB regularized | 6000 | clip+reg | 95.37%($\pm$0.12)(2) |
| FB strong reg no clip | 3000 | reg+bs32 | 91.08%($-$) |
| FB regularized double lr | 3000 | clip+reg+lr=1.6 | 95.69%($\pm$0.06)(2) |
| FB strong reg double lr | 3000 | clip+reg+bs32+lr=1.6 | 95.23%($-$) |
| FB regularized half lr | 3000 | clip+reg+lr=0.4 | 94.70%($\pm$0.17) |
| FB strong reg half lr | 3000 | clip+reg+bs32+lr=0.4 | 94.97($\pm$0.05) |

Table 5: Hyperparameter ablation studies for experiments considered in Table 1. Validation accuracies on the CIFAR-10 validation set for each experiment with data augmentations considered in Section 3. The number of trials for each experiment is included in parentheses.

| Source of Gradient Noise | Batch size | Val. Accuracy % |
|---|---|---|
| Sampling without replacement | 128 | 95.70($\pm$0.11) |
| Sampling with replacement | 128 | 95.70($\pm$0.05) |
| Sampling without replacement (fixed across epochs) | 128 | 95.25($\pm$0.07) |
| Additive $n = 0.01$ | 50'000 | 61.41($\pm$0.09) |
| Multiplicative $m = 0.01$ | 50'000 | 79.25($\pm$0.14) |
| - | 50'000 | 75.42($\pm$0.13) |

Table 6: Summary of validation accuracies on the CIFAR-10 validation dataset for *baseline* types of gradient noise in experiments with data augmentations considered in Section 3.

## C.3 NUMERICAL STABILITY

Technically, the computations in this paper may still contain residual errors in the gradient computation. This is due to use of GPUs as central computational units on which operations are non-deterministic by default (Chetlur et al., 2014; NVIDIA, 2022). We measure the amount of error between two consecutive evaluations of a single gradient in Table 7. However, the errors introduced by non-determinism are notably small in magnitude, especially compared to the effect size of noise induced by normal mini-batching (which we highlight in the first row). Even over a full training run, where the baseline SGD would take 117'000 steps with this magnitude of gradient noise (totaling an upper bound of $\approx 5534$ points of relative gradient error), these errors estimates would remain small as only 3000 steps are taken (totaling an upper bound of $\approx 2.77\mathrm{e}{-4}$ units of relative gradient error). We note that we implement the accumulation of gradients over the whole dataset with a numerically stable online mean, instead of relying on the naive accumulation in gradient leafs as in default `pytorch`. We also experimented with accumulating into higher precision (double), but did not find any effects in comparison to accumulation in single precision.

We verify the minimal impact of these findings through a sanity check: We repeat experiments from Table 1 in Table 8 in the slow deterministic mode of `cudnn` (and further disable all possibly non-deterministic operations in `pytorch`), but find no significant differences. We also check for confounding effects introduced by single floating point precision used in our experiments by re-running experiments in double precision. Both variations fail to find significant differences to the non-deterministic, single precision experiments that form the remainder of this work.

## C.4 OTHER TECHNIQUES FOR GENERALIZATION

Several other strategies to increase generalization performance have been proposed in literature about large-batch training. Here we enumerate some of the strategies as alternative to gradient clipping in the main body. The baseline here is 87.36%($\pm$1.23), i.e. FB with long training. Label

| Experiment | Batching | Modifications | Total Error | Relative Error |
|---|---|---|---|---|
| Baseline SGD | ✓ | - | 0.0988 ( ± 0.0055) | 0.0473(±0.01074) |
| Baseline FB | ✗ | - | 6.57e−7 ( ± 4.20e−8 ) | 3.40e−7 ( ± 7.30e−8 ) |
| FB clipped | ✗ | clip | 8.93e−8 ( ± 3.05e−8 ) | 3.57e−7 ( ± 1.22e−7 ) |
| FB regularized | ✗ | clip+reg | 1.15e−7 ( ± 2.21e−8 ) | 4.60e−7 ( ± 8.83e−7 ) |
| FB strong reg. | ✗ | clip+reg+bs32 | 9.24e−8 ( ± 3.97e−9 ) | 3.70e−7 ( ± 1.59e−8 ) |
| FB strong reg. (det.) | ✗ | clip+reg+bs32 | 0(±0) | 0 ( ± 0 ) |
| FB strong reg. (fp64) | ✗ | clip+reg+bs32 | 6.25e−13 ( ± 8.58e−13 ) | 2.50e−12 ( ± 3.43e−12 ) |

Table 7: Numerical Stability of Gradient Computations in several settings for a single sample of data, controlling for data augmentations. Shown are total and relative euclidean error averaged over 5 gradient computations on a randomly initialized model. For baseline SGD this is the gradient noise between two randomly sampled mini-batches at batch size 128 plus numerical effects. For fullbatch gradient descent this is noise produced solely by numerical approximation due to non-determinism when computing the gradient by accumulation over the entire dataset.

| Experiment | Mini-batching | Epochs | Steps | Modifications | Val. Acc.% |
|---|---|---|---|---|---|
| Baseline SGD (det.) | ✓ | 300 | 117,000 | - | 95.66(±0.09) |
| FB regularized (det.) | ✗ | 3000 | 3000 | clip+reg | 95.40(±0.08) |
| FB strong reg. (det.) | ✗ | 3000 | 3000 | clip+reg+bs32 | 95.66(±0.10) |
| Baseline SGD (fp64) | ✓ | 300 | 117,000 | - | 95.66(±0.07) |
| FB regularized (fp64) | ✗ | 3000 | 3000 | clip+reg | 95.54(±0.15) |
| FB strong reg. (fp64) | ✗ | 3000 | 3000 | clip+reg+bs32 | 95.62(±0.00) |

Table 8: Validation accuracies for each experiment considered in Section 3 for deterministic runs (referring to disabled `cudnn` non-determinism) and runs in double (`fp64`) floating point precision. All validation accuracies are averaged over 2-5 runs. These experiments show no significant difference to the non-deterministic runs in single floating point precision.

smoothing with smoothing value of 0.1 leads to a full-batch performance of 85.94%(±8.80) (which is lower on average, due to one run at 70%. If that run were treated as outlier and removed, we would find 90.34%(±0.23)). This is still significantly lower than the gradient clipping value of 93.85%(±0.10) with which label smoothing does not stack. Applying weight decay only to linear (convolutional and fully-connected) layers leads to a performance of 85.99%(±3.48). Sharpness-aware minimization (SAM) on the full gradient level leads to 68.77%(±17.71), even with our increased budget to 3000 iterations and gentle learning rate scheduling, however note the connection of gradient regularization and SAM when accumulated over mini-batches discussed in Appendix C.7.

Furthermore, regarding discussions about reducing the batch size $|B|$ for SGD, we find a validation accuracy of 94.99(±0.19) at $|B| = 32$ without hyperparameter adaptation. When the learning rate is scaled to 0.05, SGD reaches 95.56(±0.04), which is slightly below the value for $|B| = 128$. Using $|B| = 32$ for accumulation is further not an advantage for variants without regularization, *FB train longer* in Table 1 reaches 77.42% accuracy with this batch size, and *FB clipped* reaches 93.43%. Both numbers are below the performance at accumulation batch size 128.

## C.5  TRAINING WITHOUT SPIKES

Figure 5 shows training curves for stable full-batch training without spiking behavior. However, the optimization does not reach levels of performance shown for $\tau = 0.4$ in the main body within the allotted 3000 steps.

## C.6  HESSIAN-VECTOR PRODUCT APPROXIMATION

The Hessian-vector product necessary to compute the gradient of the gradient regularization of Equation (6) can also be compute by automatic differentiation instead of using finite differences. Using Pytorch's automatic differentiation via `autograd.grad` lead to a validation performance of 94.34%(−) for a preliminary experimental setup with label smoothing and $\alpha = 0.1$, for which the forward differences approximation we otherwise employ found 94.20%(−), so that both approaches performed near identical.

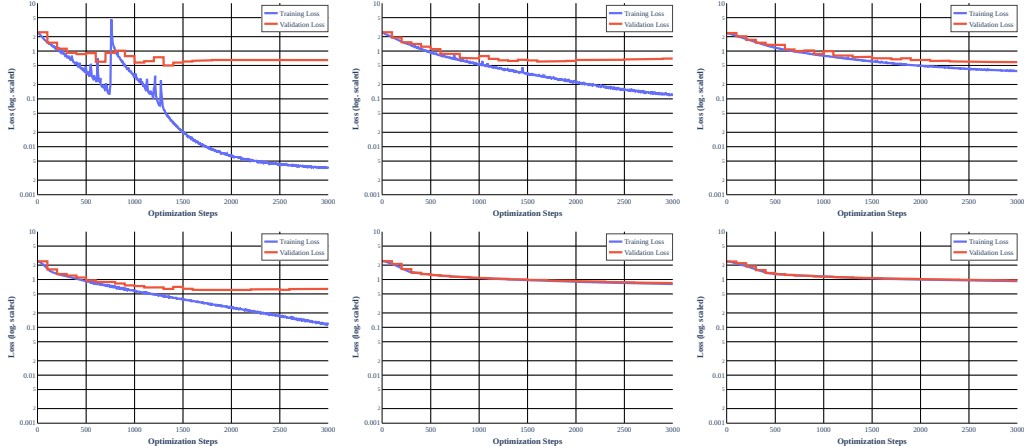

Figure 5: Cross-Entropy Loss on the training and validation set during training for full-batch gradient descent in direct comparison to Figure 2 in the main body - but with reduced learning rates. **Top Row:** $\tau = \{0.05, 0.01, 0.005\}$. **Bottom Row:** Same step sizes but with gradient clipping of $0.25$. $\tau = 0.4$ is pictured with and without clipping in the main body. Training behavior is stabilized at lower learning rates, but significantly slowed in most cases. In the first case, some overfitting appears in the validation loss, possibly corresponding to the reduced regularization of the full gradient, i.e. Barrett & Dherin (2020) and catapult behavior of Lewkowycz et al. (2020). Weight decay regularization not pictured. Validation computed every 100 steps.

Another alternative would be to improve the precision of the finite-differences approximation. Although we employ a forward-difference scheme, it is also possible to utilize a central-differences scheme which has beneficial approximation properties. Using central differences instead of central differences leads to $94.80(-)$ in another preliminary *FB regularized* setting in which forward-differences yielded $94.70(\pm 0.17)$, i.e. similar performance.

However, the additional computational effort is significant is both cases. Computing the *FB regularized* experiment on $2 \times 4$ GPUs takes about 3 hours, 14 minutes. The same experiment with central differences takes about 4 hours, 27 minutes. Finally, with automatic differentiation this takes about 8 hours and 40 minutes so that we employ the forward differences approximation only, especially in preparation for the $10\times$ and $40\times$ CIFAR-10 experiments.

### C.7 RELATIONSHIP OF HVP APPROXIMATION AND SHARPNESS-AWARE MINIMIZATION

The gradient regularization of Smith et al. (2020b) is related to the sharpness-aware minimization of Foret et al. (2021) if the latter is computed on a mini-batch level and accumulated over the entire dataset. This relationship is especially apparent when using the approximation of the Hessian-vector product with step size selection as proposed in Liu et al. (2018b). In our notation, the sharpness-aware minimization update consists of an update step based on the gradient

$$g_{\text{SAM}} = \nabla\mathcal{L}\left(\theta + \frac{\rho}{||\nabla\mathcal{L}(\theta)||}\nabla\mathcal{L}(\theta)\right). \tag{10}$$

In comparison the update via loss gradient plus derivative of the regularizer can be be written as

$$g_{\text{gradreg}} = \nabla\mathcal{L}(\theta) + \frac{\alpha\tau}{2}\frac{\nabla\mathcal{L}\left(\theta + \varepsilon\nabla\mathcal{L}(\theta)\right) - \nabla\mathcal{L}(\theta)}{\varepsilon}. \tag{11}$$

If we consider the differential step size of $\varepsilon = 0.01/||\nabla\mathcal{L}(\theta)||$ (Liu et al., 2018b) and identify $\rho = 0.01$, the we can rewrite this update to

$$g_{\text{gradreg}} = \left(1 - \frac{\alpha\tau}{2}\frac{||\nabla\mathcal{L}(\theta)||}{\rho}\right)\nabla\mathcal{L}(\theta) + \frac{\alpha\tau||\nabla\mathcal{L}(\theta)||}{2\rho}\nabla\mathcal{L}\left(\theta + \frac{\rho}{||\nabla\mathcal{L}(\theta)||}\nabla\mathcal{L}(\theta)\right). \tag{12}$$

This shows that from the point of view of Foret et al. (2021), gradient regularization is an interpolation between the normal loss gradient and the adversarial gradient that depends on the step size.

From the point of view of Smith et al. (2020b), SAM minimization accumulated over mini-batches is a finite-difference approximation of the gradient regularization for a fixed step size. Both are equivalent iff $\frac{\alpha\tau}{2}\frac{||\nabla\mathcal{L}(\theta)||}{\rho} = 1$. If $\rho = 0.01$, $\alpha = 0.5$, $\tau = 0.1$ (the hyperparameters at the end of training in our experiments), this happens whenever $||\nabla\mathcal{L}(\theta)|| = 0.2$. At the beginning of training, i.e. $\tau = 0.4$, equivalence is reached at $||\nabla\mathcal{L}(\theta)|| = 0.05$. If the gradient norm is greater than this equivalence, then the adversarial gradient dominates, if it is smaller the loss gradient $\nabla\mathcal{L}(\theta)$ dominates. However we note that according to the experiments in Appendix C.6, gradient regularization can also be implemented via automatic differentiation, so that (in the spirit of this work) the finite differences approximation itself is not necessary for the generalization effect of this regularizer.

### C.8 CHAOS THEORY - WHAT OUR RESULTS DO NOT SHOW

We would like to point out that while our results show that stochastic mini-batching (or even non-stochastic minibatching) in gradient descent is not necessary to achieve state-of-the-art generalization behavior, this does not entirely rule out stochastic modeling of the behavior of GD for deep neural networks as proposed in works such as Chaudhari & Soatto (2018); Kunin et al. (2021) and Simsekli et al. (2019). Even a full-batch gradient descent algorithm could potentially exhibit chaotic behavior on the loss surface of deep neural networks (Kong & Tao, 2020), which could be modelled by statistical techniques. In this work, we can make no statement about whether chaotic behavior exists for these examples of gradient descent and whether it has an impact on model performance.

## D ADDITIONAL DETAILS

### D.1 ASSET LICENCES

We use only CIFAR-10 data (Krizhevsky, 2009) in our experiments, for more information refer to https://www.cs.toronto.edu/~kriz/cifar.html. Code licenses for submodules are included within their respective files and can be found as part of our code release.

