# OpenReview forum: "Stochastic Training is Not Necessary for Generalization"
_ICLR.cc/2022/Conference — ICLR 2022 Poster_

### Official Review · Reviewer_o29S · 2021-10-26

**Correctness:** 4
**Technical Novelty And Significance:** 3
**Empirical Novelty And Significance:** 4
**Recommendation:** 6
**Confidence:** 3

**Main Review:**

I think the paper reveals an important message to the field. Recent works on SGD are more or less all based on the idea that minibatch training improves generalization and is worth studying on its own, and this work makes the surprising discovery that the minibatch sampling process is, in fact, unnecessary for a good generalization. The same effect can be reproduced by a series of deterministic tricks on GD

However, I feel that the way the authors present the conclusion is a little too negative against SGD. In fact, from a practitioner's point of view, I become much more convinced of the advantage of minibatch training after reading the paper (instead of finding it unnecessary). The reason is this: the paper essentially shows the following:
 - GD + minibatch sampling = GD + learning rate schedule + large learning rate + gradient clipping + gradient penalty

Namely, disregarding the computational efficiency that the minibatch technique brings, simply using mini batching alone already gives me the advantage of all the later four tricks taken together! Not to mention that a simpler technique tends to be more robust in more complicated settings.

Therefore, I think one additional conclusion that should be reached is that the minibatch sampling technique in SGD is indeed a really good technique -- but this point is not pointed out in the paper, and missing this point I think can be highly misleading for the general audience, given that the paper mainly has a negative tone against SGD. I thus think the authors need to revise their conclusion to reflect this point (or convince me why this additional conclusion is incorrect).

Additionally, I think that one crucial experiment is lacking: if one adds minibatch sampling on top of GD + learning rate schedule + large learning rate + gradient clipping + gradient penalty, what will happen? The authors are obliged to show that when the minibatch sampling is added, there is no further improvement




**Summary Of The Paper:**

The paper shows what the title says; to paraphrase, the paper shows that GD with a bunch of non-stochastic tricks can be as good as SGD with minibatch noise

**Summary Of The Review:**

Update: I am satisfied with the author's reply. I think the message is sufficiently novel and important to be published at ICLR. Though I actually would like to give a 7, due to ICLR's scoring constraints, I can only give a 6. I do not feel that giving 8 is appropriate because this paper only suggests the possibility of SGD noise being non-essential, but does not give a definitive answer.

To be specific, the main speculation/conjecture is only validated for a single example, it is completely unclear whether these results are generalizable or not. While I do find meaning and insight in the single example the authors give, I feel that more convincing empirical evidence is required to obtain a score higher than 7 and, more importantly, to reduce the speculative nature of the present conclusion.

--------------

I lean towards acceptance because I find the discovery surprising and important. However, I give a 5 because I feel that the authors missed one crucial conclusion, which might mislead the readers, and missed one crucial control experiment. I will change the score if these two points are addressed satisfactorily

---

> ### Author Response · Authors · 2021-11-18
> **Reply to Reviewer o29S**
>
> Thank you for your interesting feedback. We  address your concerns below.
>
> > However, I feel that the way the authors present the conclusion is a little too negative against SGD. In fact, from a practitioner's point of view, I become much more convinced of the advantage of minibatch training after reading the paper (instead of finding it unnecessary). [...]
> Namely, disregarding the computational efficiency that the minibatch technique brings, simply using mini batching alone already gives me the advantage of all the later four tricks taken together! Not to mention that a simpler technique tends to be more robust in more complicated settings.
>
> > Therefore, I think one additional conclusion that should be reached is that the minibatch sampling technique in SGD is indeed a really good technique -- but this point is not pointed out in the paper, and missing this point I think can be highly misleading for the general audience, given that the paper mainly has a negative tone against SGD. I thus think the authors need to revise their conclusion to reflect this point (or convince me why this additional conclusion is incorrect).
>
>
> After reading our paper with your comments in mind, we completely agree.  When we wrote the paper, it was certainly not our intent to imply that GD is preferable to SGD in practical applications, but we now want to be sure that we have not been overzealous to the point that a reader (especially a time limited reader) thinks that we are advocating that full-batch training is competitive with SGD as a practical optimizer.
> We have updated both the intro and conclusion.  We now state near the start of the paper that  “Our focus is not on fast runtimes or ultra-efficient parallelism, but rather on the implications of our experiments for deep learning theory. In fact, the extremely high cost of each full-batch update makes GD far less efficient than a conventional SGD training loop.”  We have also revised discussion in the conclusion of the paper to make this clear.  Here is an excerpt from our revised discussion: “our training routine is highly inefficient compared to SGD (taking far longer run time), and stochastic optimization remains a great practical choice for engineers in most settings.”
>
> > Additionally, I think that one crucial experiment is lacking: if one adds minibatch sampling on top of GD + learning rate schedule + large learning rate + gradient clipping + gradient penalty, what will happen? The authors are obliged to show that when the minibatch sampling is added, there is no further improvement
>
> We have added this missing experiment to the appendix. SGD performs 0.1%(+-0.18%) better with the regularization than without it, averaged over 5 trials - making it difficult to verify whether the regularization has any statistically significant benefit over base SGD at all. This is in line with recent theory which argues that this regularizer is already present in SGD.

---

> > ### Comment · Reviewer_o29S · 2021-11-19
> > **additional comments**
> >
> > Thanks for the reply and for the additional experiment.
> >
> > I have the following two additional questions/criticisms. First of all, I wondered if the same procedure can be used to explain how SGD works on a simpler network and a simpler task, for example, a simple two-layer network or a multilayer network on simple Gaussian data? The reason I want to see this is because it is important for the authors to comment on how generalizable the discovery is. Does the result only apply to ResNet + CIFAR? Or it also may be extended to other tasks? This would add (maybe a little) to the generality of the discovery.
> >
> > Secondly, after rereading the abstract and the conclusion, I find the final sentence of the paper troublesome: "If stochastic regularization does indeed have benefits that cannot be captured through non-stochastic, regularized training, those benefits are just the cherry on top of a large and complex cake." This sentence implies the following claim: "non-stochastic regularized training is sufficient to explain stochastic regularization." However, this is a severe overclaim and not supported by the paper. The paper only shows that "__it is possible__ that non-stochastic regularized training can explain stochastic regularization." I ​therefore think that the authors need to rewrite this sentence.
> >
> > The same problem appears in the sentence in the abstract: "Our observations indicate that the perceived difficulty of full-batch training __is largely the result of__ its optimization properties and the disproportionate time and effort spent by the ML community tuning optimizers and hyperparameters for small-batch training."  The statement "is largely the result" of is a severe overclaim. The paper only shows that "may be a result of." Note the huge difference here: the original claim is only supported if, for the majority of the cases (all types of architectures,tasks etc.), the claimed fact is true -- but the authors only showed one special example the claim is true. The author needs to rewrite these claims.
> >
> > I will consider changing score if the overclaiming issue is addressed.

---

> > > ### Author Response · Authors · 2021-11-19
> > > **Response to additional comments**
> > >
> > > > I have the following two additional questions/criticisms. First of all, I wondered if the same procedure can be used to explain how SGD works on a simpler network and a simpler task, for example, a simple two-layer network or a multilayer network on simple Gaussian data? The reason I want to see this is because it is important for the authors to comment on how generalizable the discovery is. Does the result only apply to ResNet + CIFAR? Or it also may be extended to other tasks? This would add (maybe a little) to the generality of the discovery.
> > >
> > > Other theoretical work (e.g. Amir et al. 2021) has shown that there exist worst-case scenarios where SGD performs better than GD provably. Our work can be seen as an investigation whether these worst-case scenarios are relevant to settings we encounter in practice.  These existing theoretical works demonstrate that analysis comparing SGD to GD on toy problems can yield conclusions which do not hold up in practice.  Therefore, we think that examining two-layer networks on Gaussian data is orthogonal to the practical questions our paper seeks to address.
> > >
> > > Nonetheless, we realize that we did not clearly explain this in our work, so we have updated our draft accordingly. In particular we have amended both statements you have identified, and we thank you for pointing this out.

---

> > > > ### Comment · Reviewer_o29S · 2021-11-21
> > > > **update**
> > > >
> > > > Thanks. I have updated the score to weak acceptance. Please check the updated summary of the review section for my reasoning about my final assessment.

---

### Official Review · Reviewer_Tx6S · 2021-10-29

**Correctness:** 2
**Technical Novelty And Significance:** 2
**Empirical Novelty And Significance:** 2
**Recommendation:** 5
**Confidence:** 3

**Main Review:**

The theoretical implications of the experiments are not significant. SGD is an empirical regularization technique.  So models trained with SGD *may* empirically overfit less than models trained in other ways.  However, SGD is neither necessary nor sufficient for models to generalize.

Some theory papers argue that SGD is sufficient for generalization.  For example, the paper, Stability and Generalization, Bousquet and Elisseeff,  shows that a technical form of stability is sufficient for generalization.  The paper, Train faster, generalize better: Stability of stochastic gradient descent, Moritz Hardt,  Benjamin Recht, Yoram Singer, argues that SGD introduces stability.  However, no one has argued that SGD is *theoretically necessary* for generalization.

The paper offers "A MORE SUBTLE HYPOTHESIS *we can modify and tune optimization hyperparameters for GD and also add an explicit regularizer in order to recover SGD’s generalization performance without injecting any noise into training.*"

To me this is an empirical technique, not a theory hypothesis.  A theory hypothesis would be something along the lines of : perform this type of regularization, and we can prove a bound on the generalization gap of the model.  An empirical theory hypothesis would be something like: perform this type of regularization, and in practise, we obtain similar generalization performance to SGD.  However, this would require more substantial empirical evidence, than is presented in the paper.



**Summary Of The Paper:**

This paper shows that it is possible to train CIFAR 10 models with full batches, and still obtain get test accuracy.  While training with SGD leads to accuracy 95+ percent, training with baseline full batch degrades accuracy in a range of 30-77 percent.  However, introducing a number of regularization discussed in section 3 can close the validation accuracy gap between SGD and FB.

This paper spends about half the time discussing theory heuristics for SGD regularization, and half the time discussion regularization heuristics which are compatible with full batch optimization.

As an empirical paper, there are a number of regularization techniques discussed in section 3: baseline sgd, stabilizing training, finite difference regularization of the Hessian of the loss equation (7), learning rate schedules, gradient clipping, gradient penalty, data augmentation. However, CIFAR-10 is no longer a challenging dataset.

From the paper: "A number of authors have studied relatively large batch training, often finding trade-offs between batch size and model performance (Yamazaki et al., 2019; Mikami et al., 2019; You et al., 2020). However, the goal of these studies has been first and foremost to accelerate training speed (Goyal et al., 2018; Jia et al., 2018), with maintaining accuracy as a secondary goal. In this study, we seek to achieve high performance on full-batch training at all costs. Our focus is not on fast runtimes or ultra-efficient parallelism, but rather on the implications of our experiments for deep learning theory."

**Summary Of The Review:**

The achievement of the paper is an empirical result: good validation accuracy on CIFAR 10 using full batch training and a number of regularization techniques.  But the paper makes a claim of implications for deep learning theory which is not substantiated.

---

> ### Author Response · Authors · 2021-11-18
> **Response to Reviewer Tx6S**
>
> > Some theory papers argue that SGD is sufficient for generalization. For example, the paper, Stability and Generalization, Bousquet and Elisseeff, shows that a technical form of stability is sufficient for generalization. The paper, Train faster, generalize better: Stability of stochastic gradient descent, Moritz Hardt, Benjamin Recht, Yoram Singer, argues that SGD introduces stability. However, no one has argued that SGD is theoretically necessary for generalization.
>
>
> We agree with the reviewer that stochasticity is not necessary for generalization.  If the reviewer already agrees with this statement, then we understand why this result may not seem too striking.  However, there are many people in the community who *do* believe that the implicit regularization of SGD is fundamental to generalization, and these are the readers who our paper is intended to reach.  In fact, we hear it often discussed among researchers that stochastic regularization is the major force behind generalization. While many theoreticians are less forceful about this issue in papers than they are in discussions, there are a number of theoretical papers that make this assertion with varying degrees of strength, and we feel it is a goal of many theoreticians to prove the connection between the implicit regularization of SGD and generalization.
>
> Some recent examples of very recent statements of this sort:
> * Amir et al. 2021a, “Never Go Full Batch (in Stochastic Convex Optimization)”
> * Amir et al. 2021b “SGD Generalizes Better Than GD (And Regularization Doesn't Help)”
> * Dauber et al. 2020 “Can Implicit Bias Explain Generalization? Stochastic Convex Optimization as a Case Study”
> * HaoChen et al. 2020 “Shape Matters: Understanding the Implicit Bias of the Noise Covariance”
>
> and some examples of ideas from the wider community about SGD:
> * https://datascience.stackexchange.com/questions/16807/why-mini-batch-size-is-better-than-one-single-batch-with-all-training-data/16818#16818
> * https://twitter.com/ylecun/status/989610208497360896
> * https://stats.stackexchange.com/questions/49528/batch-gradient-descent-versus-stochastic-gradient-descent

---

> > ### Comment · Reviewer_Tx6S · 2021-11-22
> > **updated recommendation**
> >
> > I'm raising my score - my opinion of the paper is the same, but I'm convinced that there is a large audience of people who need to know that "Stochastic Training is Not Necessary for Generalization"

---

### Official Review · Reviewer_UbWk · 2021-11-01

**Correctness:** 3
**Technical Novelty And Significance:** 3
**Empirical Novelty And Significance:** 3
**Recommendation:** 8
**Confidence:** 4

**Main Review:**

In the deep learning community, it is widely *believed* that the noise in SGD is a necessary component of any performant neural network. Over the past few years, theory papers have emerged that explicitly attribute the success of neural networks to the special properties of SGD. As most of the effort in the community has been focused on models trained with SGD-type algorithms, the conclusions of these papers so far have been mostly unchallenged. By showing that simple full-batch gradient descent can indeed achieve results similar to SGD, the paper provides an impactful contribution to the theory community. The paper is well-written and the experimental set-up is carefully designed and communicated in detailed. The fact that the authors can replicate their findings across a number of different architectures (Table 2) increases my confidence in the presented results.

I list my questions / concerns below:

* One can attribute variations in the validation accuracy to either issues in the optimization or generalization aspects of the model. When going through Tables 1 & 2 and Appendix C.2, it was not clear to me where exactly the improvements are coming from. I strongly suggest reporting metrics such as training loss & accuracy with the validation metrics in order to avoid confusion.

* I wonder if adding the explicit regularization provides the same kind of regularization as the noise in gradient descent. In particular, do SGD and GD models make similar predictions (and similar mistakes)? Will adding the explicit regularization on top of SGD improve the model?

* Figure 2 compares the loss flatness across a random direction in parameter space for different models. Can we reliably state anything regarding the flatness of the loss landscape by using 1-D projections of a O(1M) dimensional loss manifold? Moreover, note that the curvature of these models depend on the normalization parameters of the BN layer. By playing with these normalization parameters, one can change the curvature without changing the predictions of the model. This makes the cross-model curvature comparison that Figure 2 is attempting highly unreliable.

**Summary Of The Paper:**

The paper examines the role of SGD noise in the generalization performance of neural networks. In particular, the paper examines if the generalization performance of SGD-trained models can be replicated with explicitly regularized models trained with full-batch gradient descent. The authors show that this is indeed possible for a number of different architectures (ResNet, DenseNet, VGG) trained on CIFAR-10 dataset. Based on these observations, the authors conclude that any theory that relies exclusively on stochasticity of training to explain generalization in neural networks is unlikely to capture the true phenomena responsible for the success of deep learning.

**Summary Of The Review:**

The paper provides a strong contribution to the community. The main conclusions of the paper are supported with strong empirical evidence. There are a number of improvements / clarifications that can be added to the paper (which I have listed above).

---

> ### Author Response · Authors · 2021-11-18
> **Response to Reviewer UbWk**
>
> Thank you for providing this feedback and for your support of this submission. We will address open questions below, but let us know if any parts remain unclear.
>
> > One can attribute variations in the validation accuracy to either issues in the optimization or generalization aspects of the model. When going through Tables 1 & 2 and Appendix C.2, it was not clear to me where exactly the improvements are coming from. I strongly suggest reporting metrics such as training loss & accuracy with the validation metrics in order to avoid confusion.
>
> This is a great point. We have added measurements of training loss as well as full loss (including regularization) for all considered experiments in Table 1 to our appendix in Table 4.
>
> > I wonder if adding the explicit regularization provides the same kind of regularization as the noise in gradient descent. In particular, do SGD and GD models make similar predictions (and similar mistakes)? Will adding the explicit regularization on top of SGD improve the model?
>
> SGD performs 0.1%(+-0.18%) better with the regularization than without it, averaged over 5 trials - making it difficult to verify whether the regularization has any statistically significant benefit over base SGD at all. This is in line with recent theory which argues that this regularizer is already present in SGD.
>
> > Figure 2 compares the loss flatness across a random direction in parameter space for different models. Can we reliably state anything regarding the flatness of the loss landscape by using 1-D projections of a O(1M) dimensional loss manifold? Moreover, note that the curvature of these models depend on the normalization parameters of the BN layer. By playing with these normalization parameters, one can change the curvature without changing the predictions of the model. This makes the cross-model curvature comparison that Figure 2 is attempting highly unreliable.
>
> In our submission, we apply the visualization techniques from Li et al., 2018.  In particular, we use their “filter-normalization” technique that makes visualization results invariant to the choice of batch norm parameters.   We acknowledge that there is some debate as to what is the true  “correct” way to visualize flatness, and there are limitations to dimensionality reduced visualizations.  We chose this method because it captures a notion of flatness that is known to (empirically) correlate well with test error (a correlation we observe in our work as well). We found it instructive to “see” what was going on in the loss landscape during development of this project, so we decided to pass these visualizations (and hopefully some intuition) along to the reader.

---

> > ### Comment · Reviewer_UbWk · 2021-12-01
> > **Response & Potential Experiments for the Final Version**
> >
> > Thanks for the clarifications. They indeed address my earlier concerns.
> >
> > There is one remaining question that I was hoping that the authors could address in the final version. By default, Pytorch and TF tend to use non-deterministic cuDNN ops that can inject some noise into the gradient computation. We intuitively believe that the effect of these random perturbations is negligible. It would be valuable to add some experiments to the final version to verify this intuition. In particular, it would be nice to see:
> >
> > 1. Will the results change if deterministic ops are enabled? (See [link](https://github.com/NVIDIA/framework-determinism/blob/master/pytorch.md) for some pointers).
> > 2. What is the magnitude of the gradient noise caused by the cumulative effect of non-deterministic ops and the device? For example, one can measure $\ell_{\infty}$ & $\ell_2$ norm of the difference between two identical gradient evaluations and compare them to the gradient norm itself.
> >
> > As a pointer, for CIFAR-10 models trained with float32 precision in TF, I have observed $O(10^{-6})$ $\ell_{\infty}$ variation in the past. It would be great to see if similar behaviors hold here and to verify that their effect is limited.

---

> > > ### Author Response · Authors · 2021-12-01
> > > **Precision Measurements**
> > >
> > > This is an interesting question, thank you for the feedback.
> > > We ran some experiments with cudnn determinism and double precision in earlier stages of the project (and still have these options in our code submission), but did not find a noticable difference between non-deterministic/deterministic and float32/float64 performance. We'll rerun these experiments in more detail for the next update of our submission and we will measure the floating point precision of our implementation as suggested.

---

### Official Review · Reviewer_XRTg · 2021-11-02

**Correctness:** 4
**Technical Novelty And Significance:** 4
**Empirical Novelty And Significance:** 4
**Recommendation:** 10
**Confidence:** 5

**Main Review:**

== Updated score ==

I am satisfied by the results of the control experiments, and the explanations of the authors. I have read the other reviewer comments, and I believe the work will be quite valuable for the ICLR community and thus have changed the score to a strong accept.

========

Authors do thorough experiments to bring out the nuances around the topic with the CIFAR-10 dataset, and Residual Networks.

Strengths:
+ Very thorough set of experiments ablating over various sources of stochasticity
+ Detailed explanation of experimental protocol and reproducibility
+ Shows explicit regularization can bridge the gap, and shows efficient implementation and also discusses prior work (SAM)

Weakness/Questions:
+ Does explicit regularization help SGD (stochastic mini-batch training) (This is an important control experiment)? If it changes the target solution quality - how does one reason about matching generalization of stochastic mini-batch training.
+ Do the results hold across wider range of datasets, While I can see how much larger datasets could become more expensive - it would have been very satisfying to verify.

Some other comments: Relevant citations that are missing, which hopefully authors can easily address:
Second order optimization that seem to work well for Neural Nets:
https://arxiv.org/abs/1503.05671 (KFAC)
https://arxiv.org/abs/2002.09018 (Shampoo)
https://www.bmvc2020-conference.com/assets/papers/0479.pdf (L-BFGS)

Recent work on curvature and gradient clipping (and other techniques):
https://arxiv.org/pdf/2110.04369.pdf



**Summary Of The Paper:**

Paper presents empirical results that show that full-batch training of neural networks can still generalize with appropriate explicit regularization. Authors propose explicit regularization to minimum norm solution and discusses techniques for efficient implementation as well as relation to prior work.

**Summary Of The Review:**

This paper provides convincing contrary evidence that shows stochastic mini-batching by itself is not unique, and can be substituted with explicit regularization, and techniques such as gradient clipping.

---

> ### Author Response · Authors · 2021-11-18
> **Response to Reviewer XRTg**
>
> Thank you for your feedback and interest in this work, we answer your questions below, but let us know if further questions remain:
>
> > Does explicit regularization help SGD (stochastic mini-batch training) (This is an important control experiment)? If it changes the target solution quality - how does one reason about matching generalization of stochastic mini-batch training.
>
> Thank you for pointing this out to us. We have added the numbers for this experiment  to our ablation in the appendix. SGD performs 0.1%(+-0.18%) better with the regularization than without it, averaged over 5 trials - making it difficult to verify whether the regularization has any statistically significant benefit over base SGD at all. This is in line with recent theory which argues that this regularizer is already present in SGD.
>
> > Do the results hold across wider range of datasets, While I can see how much larger datasets could become more expensive - it would have been very satisfying to verify.
>
> We have limited ourselves here to a well-studied, but small “default” dataset  which is further small enough that we can artificially enlarge it (e.g. our 10X cifar with static augmentation) as in Sec 4.  We would love to have results on larger datasets, especially ImageNet, but this would require computational resources orders of magnitude beyond what we have in our lab. Even compared to some previous work on large-batch ImageNet training like Goyal et al. 2018, the discussed strategy would incur additional costs per-step (due to regularization) and many more communication rounds due to increased numbers of steps - before even mentioning the 10X dataset size increase required for static data augmentation. It is further unclear inhowfar common acceleration techniques can be applied in the given full-batch scenario. For example, we investigated mixed-precision training in earlier stages of the project and found it to degrade performance of the finite-diff approximation of the regularizer. Adapting all techniques in this paper to efficient training for larger datasets seems to us like a good place for future work, but it may have to be taken up by an industry lab with more resources.
>
> > Some other comments: Relevant citations that are missing, which hopefully authors can easily address: Second order optimization that seem to work well for Neural Nets: https://arxiv.org/abs/1503.05671 (KFAC) https://arxiv.org/abs/2002.09018 (Shampoo) https://www.bmvc2020-conference.com/assets/papers/0479.pdf (L-BFGS)
>
> This is a good point - higher-order optimization methods have had a number of great successes using large batch sizes. We have added this to our literature review.
>
> > Recent work on curvature and gradient clipping (and other techniques): https://arxiv.org/pdf/2110.04369.pdf
>
> Thank you for pointing us to this reference.

---

### Official Review · Reviewer_g1Z8 · 2021-11-08

**Correctness:** 3
**Technical Novelty And Significance:** 3
**Empirical Novelty And Significance:** 2
**Recommendation:** 6
**Confidence:** 4

**Main Review:**

The paper has strong merits on the technical side, offering several well-executed novel experiments:
- First full-batch training regime of cifar10 with competitive validation accuracy
- Removal of any noise affecting training (full-batch, fixed batch normalization slices)
- "Augmentation aware" full-batch training -- by pre-enlarging the dataset with fixed augmentation, thus making sure comparison to practice is fair.

The authors also offer some practical methods that may prove useful in the future, by offering gradient-clipping and regularization to combat large batch induced accuracy degradation and failures.
The paper is also well written and easy to follow with a good cover of past efforts on this topic.

My main concern with the work is that several choices seem arbitrary and not well explained:
- Number of iterations: as seen in past works (Hoffer et al. 17',  Shallue et al. 19'), adapting the training regime with more training steps can close the gap between large and small batch training. This is also apparent in this work, as authors increase the number of steps to 3000 (10x the number of epochs) to reach the comparable accuracy milestone. However, this number is not justified in any way and no further exploration in that regard is made. Some possible questions that arise: (1) can we apply the same number of iterations as in small batch (117K) to close the gap completely with no need for additional regularization? (2) what is the tradeoff in terms of accuracy? (3) in terms of computational effort?
- Regularization - the authors choose a specific regularizer to replace the implicit bias offered by SGD. However, several other choices can be offered instead (e.g SAM, preconditioning). More importantly, does the regularization really replaces SGD? what accuracy do we get when the same regularizer is applied to baseline SGD training?
- Learning rate adjustment - authors use x2 learning rate instead of previous scaling methods (sqrt/linear scaling), in what seems like a completely arbitrary choice. As previous works used signal-to-noise ratio of gradients to suggest scaling of the learning rate, I expect authors to suggest and demonstrate why (and if) these break at the full-batch limit, when no noise is introduced.


**Summary Of The Paper:**

The paper discusses large batch training at its limit -- full-batch cifar10 training when no stochasticity is introduced. This setting allows authors to examine and discuss the common conception about the generalization benefits of SGD. The authors show that replacing implicit bias of SGD with explicit regularization can eliminate the generalization gap.

**Summary Of The Review:**

As explained in the review, the paper has several technical merits and interesting experiments but is clouded with ill-justified and what seems like arbitrary choices and solutions. Currently, I feel these hurt the potential value of the paper to the field.

My view is that the paper in its current form should be rejected, but I suggest the following improvements:
- Discuss regime adaptation (increasing number of steps) similar to Hoffer et al. by performing additional experiments with a varying number of epochs. I understand that a complete 117K steps training in full-batch is too cumbersome, but the current single datapoint of 3000 epochs seems arbitrary and not justified in any way.
- Regularization on baseline SGD -- does it help? will be very interested to see a discussion either way.
- LR adjustment and gradient clipping justification -- perhaps additional measurements of gradient noise may show why these are needed over "traditional" scaling techniques.

If authors work to lift these concerns in their rebuttal, I will consider raising my score.



----- Post rebuttal update -----
The authors answered most of my concerns, so I'm updating my score as promised. I still feel some of the choices are arbitrary (or at least not substantiated enough) and may apply to the specific case presented only (dataset, model), so I will raise my score 5->6.

---

> ### Author Response · Authors · 2021-11-18
> **Response to Reviewer g1Z8**
>
> Thank you for your extensive feedback. We address your questions concerning ablations and hyperparameter choices below.
>
> > Discuss regime adaptation (increasing number of steps) similar to Hoffer et al. by performing additional experiments with a varying number of epochs. I understand that a complete 117K steps training in full-batch is too cumbersome, but the current single datapoint of 3000 epochs seems arbitrary and not justified in any way.
>
> Indeed, the Hoffer rule of taking 117k steps is not tractable using full-batch optimization due to excessive runtime costs.   We agree that the reader will likely be interested to see how the number of training steps impacts accuracy and whether longer training would indeed close the gap without the need for regularization.  We have now added an ablation study in Table 5 of the appendix to address this issue.  We find that running our training loop for 6000 steps does not produce any accuracy improvements over the 3000-step run.  We also include a huge run with 40K steps, which also did not succeed in closing the gap with small batch training.
> Unfortunately, it is known that the Hoffer rule (which was validated on “moderate” batch sizes) does not scale to extremely large batch sizes.  You et al. 2020 have evaluated the “train longer” paradigm of Hoffer et al., but have found it not to hold up in the extremely large batch regime. This was also observed in Li et al. 2021 (Appendix F3).   In our own experiments, we have also found the Hoffer rule to be sub-optimal in the regime of extremely large batches.
>
>
> > Regularization on baseline SGD -- does it help? will be very interested to see a discussion either way.
>
> Another great question.  We have completed a round of experiments on standard SGD with regularization, and added them to our ablations in the appendix. SGD performs 0.1%(+-0.18%) better with the regularization than without it, averaged over 5 trials - making it difficult to verify whether the regularization has any statistically significant benefit over base SGD at all. This is in line with recent theory which argues that this regularizer is already present in SGD.
>
> > Learning rate adjustment - authors use x2 learning rate instead of previous scaling methods (sqrt/linear scaling), in what seems like a completely arbitrary choice. As previous works used signal-to-noise ratio of gradients to suggest scaling of the learning rate, I expect authors to suggest and demonstrate why (and if) these break at the full-batch limit, when no noise is introduced.
>
> Our choices here are not ad-hoc, but rather they are based on previous literature. Previous scaling methods are only applicable up to a limit of critical mini-batch size, as discussed in Ma et al. 2018, Jain et al. 2019, Zhang et al. 2019. This is further investigated empirically in McCandlish et al. 2018 and Shallue et al. 2019. It is true that signal-to-noise ratio is useful for SGD, but full-batch gradient GD has no noise, and so the signal to noise ratio is infinite, causing SNR-based rules to break down. As discussed in Shallue et al. 2019, existing theory does not describe the extremely large-batch setting, and specific tuning for a given batch size (or full-batch in our setting) is required. We thus started with the learning rate of SGD (0.1) and then increased the learning rate by doubling until no improvement could be observed. We believe this to be a sufficiently strong step size selection, given that Shallue et al. 2019 also find that “the region in metaparameter space corresponding to rapid training in terms of step-count grows larger” with increased batch sizes, i.e. exact step size selection is not as crucial for extremely large batches.  To explore this issue deeper, we have added an ablation to the appendix that shows results with 0.5x and 2x the learning rate in the appendix.  We see that smaller learning rates hurt performance a bit, while the large learning rate produces similar results.
>
> > LR adjustment and gradient clipping justification -- perhaps additional measurements of gradient noise may show why these are needed over "traditional" scaling techniques.
>
> There is no gradient noise in updates of full-batch gradient descent in our final experiments. While the magnitudes of gradient noise during training with different LR and clipping would certainly be interesting to study in the large (but not close to full) batch regime, we have focused here on the full-batch case because of its theoretical ramifications. The instability of training for over-parametrized models in the full-batch setting was analyzed in works such as Cohen et al. 2020, who show that instability exists (and exists independent of gradient noise) and has to be controlled to optimally train these models.

---

### Decision · Program_Chairs · 2022-01-20

**Decision:**

Accept (Poster)

**Comment:**

This paper carefully shows how all the stochastic elements in neural network training could be removed (by using full batch, and a dataset with fixed augmentation) and still maintain good performance, by adjusting hyper-parameters and adding explicit regularization.
All reviewers were eventually positive and recommended acceptance, except one reviewer, who was initially not aware of the recent theoretical interest in this question, and was therefore less surprised.

There are three remaining issues with the current version:
1) Operations in cuDNN are, by default, non-deterministic, but can be made deterministic. Though I believe this will not affect the final results, without it, the title and conclusions of this paper are technically unjustified. The authors have agreed to add this to the camera ready version of the paper.
2) Deterministic training was only shown on CIFAR10. I understand ImageNet would be too heavy for this task, but there are many other small and medium-sized datasets, and I think showing that this on several such datasets would strengthen the message of this paper, and convince more readers.
3) The question of how to achieve good performance with deterministic training is still mostly unknown, as it seems to require significant hyperparameter search (with unknown sensitivity), and no conclusion was reached regarding the how to properly adjust them. However, I agree that a good answer to this question might not exist.

Lastly, I recommend the authors to mention in the main paper the new baseline experiment, where the explicit regularization is added to SGD. I saw it in the appendix, but I didn't see it mentioned in the main paper (maybe I missed it). I think without it, many readers will not be fully convinced (as several reviewers requested it).

---

> ### Public Comment · ~Jonas_Geiping1 · 2022-03-16
> **Final Version**
>
> We now uploaded the final version and (for anyone looking for answers in the future) we note that questions about non-determinism are now answered in Appendix C.3, additional discussion about hyperparameter choices and stability is included in Appendix C.2 and the regularized baseline can be found in table 1.